# A Provable Quantile Regression Adapter via Transfer Learning

## Abstract

Adapter-tuning strategy is an efficient method in machine learning that introduces lightweight and sparse trainable parameters into a pretrained model without altering the original parameters (e.g., low-rank adaptation of large language models). Nevertheless, most existing adapter-tuning approaches are developed for risk-neutral task objectives and the study on the adaptation of risk-sensitive tasks is limited. In this paper, we propose a transfer learning-based quantile regression adapter to improve the estimation of quantile-related risks by leveraging existing pretrained models. We also establish a theoretical analysis to quantify the efficacy of our quantile regression adapter. Particularly, we introduce a transferability measure that characterizes the intrinsic similarity between the pretrained model and downstream task in order to explain when transferring knowledge can improve downstream learning. Under appropriate transferability and structural assumptions, we establish error bounds for the estimation and out-of-sample prediction quality by our quantile regression adapter. Compared to vanilla approaches without transfer learning, our method is provably more sample efficient. Extensive numerical simulations are conducted to demonstrate the superiority and robustness of our method empirically.

## 1 Introduction

Transfer learning with large pretrained models has demonstrated great successes recently (Devlin et al., 2019; Wang et al., 2019; Liu et al., 2023). The significant value of efficiently adapting large, general pretrained models to specific tasks with limited data has generated extensive interest from both researchers and practitioners (Pan & Yang, 2009; Kaplan et al., 2020; Zhuang et al., 2020; Han et al., 2021; Yuan et al., 2020; Ding et al., 2023; Wu et al., 2023; Chen et al., 2024). However, adapting the large models can be expensive. For example, transformer-based language models like BERT have around 340 million parameters (Devlin et al., 2019), and GPT-2 has around 1.5 billion parameters (Radford et al., 2019). Adapting all these parameters is prohibitively costly and even practically infeasible.

One popular transfer learning approach is adapter-tuning strategy, which leverages knowledge from pretrained model in a parameter-efficient manner—instead of directly fine-tuning all original parameters of the pretrained model, the adapter-tuning strategy introduces lightweight and sparse parameter modules to the pretrained model and only optimizes these modules without altering the original parameters during fine-tuning. This design offers two key advantages. First, it provides better accessibility by reducing the computational demands, as fine-tuning large pretrained models from scratch requires vast resources and excessive data. Second, the newly introduced parameter modules can flexibly learn target representations while preserving knowledge from the source domain, avoiding catastrophic forgetting. Previous works have shown that the adapter-tuning strategy achieves effective and computationally economical performance across various downstream tasks (Rebuffi et al., 2017; Hu et al., 2022; Wang & Liang, 2024; Raffel et al., 2020; Wu et al., 2024).

Despite the seemingly broad applicability of adapter-tuning, most existing approaches focus on risk-neutral task objectives, and research on the adaptation for risk-sensitive tasks is limited. These specific downstream tasks are ubiquitous and often critical in practice. For example, in financial risk management, institutions are concerned with the occurrence of rare, extreme situations in order to ensure sufficient capital reserves (Maiti, 2021; Ayse Demir & Murinde, 2022). In healthcare

management, identifying patients with high risk for certain conditions is crucial for early diagnosis and timely intervention. (Chen et al., 2014; Wei et al., 2019; Aktar et al., 2023). Similarly, one of the primary goals in climate and disaster studies is predicting extreme weather events, such as unprecedented temperatures or precipitation (Cai & Reeve, 2013; Naess et al., 2013). Although many existing transfer learning methods aids in predicting averaged risk of these events, the importance of tail probabilities suggest that the predominant risk-neutral learning objectives might not be adequate.

To address this problem, we investigate the transfer of knowledge in quantile regression, a widely used model that predicts the conditional quantiles of a variable of interest given fixed contextual information (Koenker & Hallock, 2001). Compared to the ordinary least squares (OLS) which focuses on predicting conditional mean values, quantile regression offers greater flexibility in examining different parts of the outcome distribution, thereby enabling the risk-sensitive prediction of extreme events. We focus on the following research question:

*Is it possible to design a provably effective transfer learning algorithm for quantile regression?*

In this paper, we aim to design a quantile regression adapter that leverages the knowledge of the pretrained models to enhance the performance of adaptation while maintaining the computational efficiency.

Inspired by the adapter-tuning strategy, we propose a quantile regression adapter that injects task-specific parameters into a pretrained model. The task-specific parameters are trained through the empirical quantile loss minimization along with a regularization penalty. The penalty term can be selected as certain vector or matrix norm in order to maintain a sparse or low-rank structure of additional parameters. Note that our method can naturally extend beyond vector/matrix-based parameters to deep neural networks by imposing a low-rank decomposed structure of networks, following the same principal of low-rank adaptation as in large language models. In this case, the size of trainable task-specific parameters can drop even more significantly (Hu et al., 2022; Zhang et al., 2023; He et al., 2023; Kim et al., 2024; Wang & Liang, 2024). Overall, our approach helps reduce the computational burden and memory usage in training and inference, especially when leveraging hardware acceleration (Dave et al., 2020; Reuther et al., 2020; Louizos et al., 2018), and the usage of regularization can also mitigate the risk of overfitting in the fine-tuning of downstream task using scarce data.

Our main contributions are summarized as follows.

- We propose a transfer learning algorithm to learn quantile information based on the adapter-tuning strategy. Our adapter injects additional learnable parameters of sparse or low-rank structure to the pretrained parameters in order to learn from downstream data while leveraging the knowledge of pretrained model.

- We borrow the concept of "sparsity" from high-dimensional statistics theory to explain why the knowledge can be transferred from the pretrained model. Based on this, we establish performance guarantee for our quantile regression adapter under linear structural model and quantify the improvement of our approach than vanilla learning without using pretrained knowledge.

- We evaluate the adaptation performance of our algorithm through numerical simulations on specific downstream tasks. Compared to baselines, our method achieves better performance in adaptation and exhibits robustness with heteroscedastic data.

## 1.1 RELATED WORK

**Adapter-tuning strategy**. The adapter-tuning strategy is a parameter-efficient transfer learning method that introduces new trainable modules into a pretrained model while keeping the pretrained model's original parameters unchanged. These modules are often specifically designed for computational efficiency due to excessive model size. For example, LoRA-like modules Hu et al. (2022); Wang & Liang (2024); Zhang et al. (2023); Kim et al. (2024); Luo et al. (2023) introduce a "low-rank" structure by decomposing the dense layers into low-rank matrices. Other studies apply network pruning or weight regulations to maintain "sparse" parameters (He et al., 2022; Zeng et al., 2023; Guo et al., 2021; Fu et al., 2023). More literature on adapter structure design can be found

in (Hu et al., 2023; Xu et al., 2023). These approaches offer valuable insights for designing new transfer learning algorithms for quantile regression.

**Quantile regression**. Quantile regression Koenker & Hallock (2001) is a powerful technique for estimating conditional quantile functions and is widely utilized across various fields, including economics (Bonaccorsi et al., 2020; Maiti, 2021), healthcare (Chen et al., 2014; Wei et al., 2019; Aktar et al., 2023), and management science (Ban & Rudin, 2019; Shah et al., 2023; Zhang et al., 2024). In recent years, quantile regression has served as an auxiliary or alternative objective in various machine learning tasks, such as uncertainty quantification (Romano et al., 2019; Feldman et al., 2023; Teneggi et al., 2023; Huang et al., 2024), risk-averse reinforcement learning (Dabney et al., 2018; Yang et al., 2019; Kuznetsov et al., 2020; Shi et al., 2024), and time series prediction (Wen et al., 2017; Yang et al., 2022; Eisenach et al., 2022; Kan et al., 2022). Our paper mainly focus on solving quantile regression via adapter-tuning and transfer learning. Within this stream of literature, our work is most closely related to Zhang & Zhu (2022) and Jin et al. (2023), both studying transfer learning for the linear quantile regression model. We highlight that their algorithms are not based on the adapter-tuning strategy but a pooling-then-debiasing technique and, therefore, not applicable when an existing pretrained model is available. Additionally, it is unclear how their algorithms could be generalized to nonlinear models even in conceptual.

**Statistical analysis in transfer learning.** Previous works have established statistical guarantees for transfer learning in various high-dimensional regression contexts, including linear regression (Li et al., 2022; Bastani, 2021; Mousavi Kalan et al., 2020; Lin & Reimherr, 2022), generalized linear models (Tian & Feng, 2023), non-parametric regression (Cai & Pu, 2024), and quantile regression (Zhang & Zhu, 2022; Jin et al., 2023). Unlike our methods, these studies typically assume access to both source and target data during the adaptation. They design algorithms that first pool all pretrained and target data together and then apply debiasing estimators using the target data. Alternatively, their analysis depends on specific loss objectives design used to train the pretrained model. Our theoretical analysis does not impose restrictions on the empirical loss form of the pretrained model. This flexibility is advantageous because pretrained models may use either unsupervised or supervised objectives (Devlin et al., 2019; Howard et al., 2019; Ridnik et al., 2021). Additionally, we focus on the case with only the usage of target data for task-specific module, which does not require access to source data during adaptation in downstream tasks.

## 1.2 NOTATIONS

Throughout this paper, we use bold lowercase letter to refer a vector (e.g. $\boldsymbol{x} \in \mathbb{R}^d$), and bold uppercase letters to refer a matrix (e.g., $\boldsymbol{X} \in \mathbb{R}^{d \times d}$). For an integer number $d$, $[d]$ denotes the set $\{1, 2, \cdots, d\}$. For any fixed vector $\boldsymbol{x} \in \mathbb{R}^d$, its support is the set of indices with non-zero value, i.e. $supp(\boldsymbol{x}) = \{j \subseteq [d] : x_j \neq 0\}$. Let $\mathbb{S}$ be a subset of $[d]$, $\boldsymbol{x}_{\mathbb{S}} \in \mathbb{R}^d$ denotes the vector such that $[\boldsymbol{x}_{\mathbb{S}}]_i = x_i$ if $i \in \mathbb{S}$ and $[\boldsymbol{x}_{\mathbb{S}}]_i = 0$ otherwise. The cardinality of set $\mathbb{S}$ is denoted by $|\mathbb{S}|$. Given a vector $\boldsymbol{x} \in \mathbb{R}^d$, $\|\boldsymbol{x}\|_p$ denotes the $L^p$-norm, $p \geq 1$, i.e. $\|\boldsymbol{x}\|_p = (\sum_{i=0}^d |x_i|^p)^{1/p}$ and $\|\boldsymbol{x}\|_\infty = \max_{i \leq d} |x_i|$. $\mathbf{1}_E(\cdot)$ is the indicator function, which takes value 1 when the event $E$ happens and 0 otherwise. Lastly, for a matrix $\boldsymbol{X} \in \mathbb{R}^{d \times d}$, $\|\boldsymbol{X}\|_2$ denotes its spectral norm and $\boldsymbol{X}^{1/2}$ is its matrix square root.

## 2 ALGORITHM DEVELOPMENT

### 2.1 PROBLEM SETTING

We start with a brief introduction to the quantile regression problem formulation. Given the covariate $\boldsymbol{x} \in \mathbb{R}^d$ and a scalar response $y \in \mathbb{R}$, the $\tau$-th conditional quantile function of $y$ conditional on $\boldsymbol{x}$ is defined as

$$F_{y|\boldsymbol{x}}^{-1}(\tau) = \inf\{\xi : F_{y|\boldsymbol{x}}(\xi) \geq \tau\}. \tag{1}$$

Here $F_{y|\boldsymbol{x}}(\cdot)$ is the cumulative distribution function of $y$ given $\boldsymbol{x}$ and $0 \leq \tau \leq 1$. The ordinary quantile regression model assumes that

$$F_{y|\boldsymbol{x}}^{-1}(\tau) = f(\boldsymbol{x}; \boldsymbol{\theta}^\star), \tag{2}$$

where function $f(\boldsymbol{x}; \boldsymbol{\theta})$ is a parametric function class parameterized by $\boldsymbol{\theta}$ and $\boldsymbol{\theta}^\star$ is the unknown true parameter. To train quantile regression, a standard loss function defined at population level is

$$\mathcal{R}_\tau(\boldsymbol{\theta}) = \mathbb{E}_{(\boldsymbol{x}, y) \sim p} \left[ \rho_\tau \left( y - f(\boldsymbol{x}; \boldsymbol{\theta}) \right) \right], \tag{3}$$

where $p$ is the joint distribution of $(\boldsymbol{x}, y)$ and the ordinary quantile loss (i.e., pinball loss) $\rho_\tau(\cdot)$ is defined as

$$\rho_\tau(x) = \begin{cases} \tau \left( y - f(\boldsymbol{x}; \boldsymbol{\theta}) \right), & y \geq f(\boldsymbol{x}; \boldsymbol{\theta}), \\ (1 - \tau) \left( f(\boldsymbol{x}; \boldsymbol{\theta}) - y \right), & \text{o.w.} \end{cases} \tag{4}$$

This objective utilizes an asymmetric convex loss to penalize the prediction error $y - f(\boldsymbol{x}; \boldsymbol{\theta})$. When the error is negative, the penalty is proportional to $\tau$ and otherwise, $1 - \tau$. When $\tau = 1/2$, the quantile loss becomes the median absolute deviation loss. Since the true parameter $\boldsymbol{\theta}^\star$ optimizes $\mathcal{R}_\tau(\boldsymbol{\theta})$, by minimizing the empirical version of $\mathcal{R}_\tau(\boldsymbol{\theta})$, we can obtain a good estimator of $\boldsymbol{\theta}^\star$.

Specifically, let $\mathcal{D} = \{(y_i, \boldsymbol{x}_i)\}_{i=1}^n$ be the dataset of a target downstream task, define

$$\widehat{\boldsymbol{\theta}} = \underset{\boldsymbol{\theta} \in \mathbb{R}^d}{\arg\min} \, \widehat{\mathcal{R}}_\tau(\boldsymbol{\theta}) = \frac{1}{n} \sum_{i=0}^n \rho_\tau \left( y_i - f(\boldsymbol{x}_i; \boldsymbol{\theta}) \right). \tag{5}$$

Then $\widehat{\boldsymbol{\theta}}$ is an approximation of true parameter $\boldsymbol{\theta}^\star$. When sample size $n$ increases, $\widehat{\boldsymbol{\theta}}$ converges to $\boldsymbol{\theta}^\star$ at rate of $\mathcal{O}(n^{-1/2})$ under appropriate regularity conditions.

On the other hand, in some scenarios, for a target qunatile regression task, before the empirical quantile loss is constructed, a pretrained model based on another source data may already exist. We assume that a pretrained model using source data $\mathcal{D}_s$ is obtained via

$$\widehat{\boldsymbol{\theta}}_s = \underset{\boldsymbol{\theta} \in \mathbb{R}^d}{\arg\min} \, \mathcal{L}(\boldsymbol{\theta}; \mathcal{D}_s), \tag{6}$$

where $\mathcal{D}_s$ denotes the source dataset and $\mathcal{L}(\cdot; \cdot)$ is the training loss for source task. When the pretrained model is correctly specified and the sample size of $\mathcal{D}_s$ goes up, $\widehat{\boldsymbol{\theta}}_s$ converges to

$$\boldsymbol{\theta}_s^\star = \underset{\boldsymbol{\theta} \in \mathbb{R}^d}{\arg\min} \, \mathbb{E}_{\mathcal{D}_s \sim p_s} \left[ \mathcal{L}(\boldsymbol{\theta}; \mathcal{D}_s) \right], \tag{7}$$

the minimizer of population loss defined for the source task, where $p_s$ is underlying distribution for source data. As a result, if $\boldsymbol{\theta}_s^\star$ is close to $\boldsymbol{\theta}^\star$, then target quantile training appropriately adapted from $\widehat{\boldsymbol{\theta}}_s$ may accelerate convergence and improve the performance.

## 2.2 Quantile Regression Adapter via Transfer Learning

Consider a scenario where the true parameter of the source task $\boldsymbol{\theta}_s^\star$ is close to that of target quantile regression task $\boldsymbol{\theta}^\star$. Let $\boldsymbol{\delta}^\star = \boldsymbol{\theta}^\star - \boldsymbol{\theta}_s^\star$ be the difference among two sets of true parameters, which is close to zero and sparse. If the source data $\mathcal{D}_s$ is sufficient and the pretrained model is trained well, $\boldsymbol{\theta}^\star \approx \widehat{\boldsymbol{\theta}}_s$. Then we can use parameter of format $\widehat{\boldsymbol{\theta}}_s + \boldsymbol{\delta}$ to learn $\boldsymbol{\theta}_s^\star$ as a adaptation, where the optimization is taken over $\boldsymbol{\delta}$, i.e., approximating the conditional quantile $F_{y|\boldsymbol{x}}^{-1}(\tau)$ as $f(\boldsymbol{x}, \widehat{\boldsymbol{\theta}}_s + \boldsymbol{\delta})$.

On the other hand, since the true parameter difference $\boldsymbol{\delta}^\star$ is sparse and locates near zero, instead of searching over the whole parameter space $\mathbb{R}^d$, which could be high-dimensional, we can restrict our attention in low-dimensional subspaces. Equivalently, we add a regularization term on $\boldsymbol{\delta}$ in the ordinary quantile loss to penalize its deviation from zero. Specifically, we propose the following loss function as the quantile regression adapter for target task

$$\mathcal{L}^a(\boldsymbol{\delta}; \mathcal{D}) = \frac{1}{n} \sum_{i=0}^n \rho_\tau \left( y_i - f(\boldsymbol{x}_i; \widehat{\boldsymbol{\theta}}_s + \boldsymbol{\delta}) \right) + \lambda \cdot g(\boldsymbol{\delta}), \tag{8}$$

where $\rho_\tau(\cdot)$ is the ordinary quantile loss defined in Equation 4 and $g(\cdot)$ is a regularization term for $\boldsymbol{\delta}$. Tuning parameter $\lambda$ controls the power of regularization. Regularization disencourages the target estimator from deviating from the source model $\widehat{\boldsymbol{\theta}}_s$ significantly. If the true source model is indeed close to the true target model and the pretrained model fits the true source model well, restricting the target estimator to be close to the pretrained model can provide an effective update direction for the

target task training. Since the original parameters in pretrained model is frozen during adaptation, the source knowledge keeps unchanged as well.

From the perspective of high-dimensional statistics, when the dimension of features $d$ is much larger than the sample size of target task $n$, the ordinary quantile regression can lead to inconsistent estimation of true parameter (Wainwright, 2019; Geer, 2000). This inconsistency motivates the use of penalization techniques to eliminate almost regressors whose true population coefficients are zero, making it possible to recover consistency. In Section 3, we will theoretically define and quantify the sparsity between the source model and target model, and provide a theoretical understanding of the behavior of adapter.

By choosing specific form of $f(\boldsymbol{x}, \boldsymbol{\theta})$ and $g(\boldsymbol{\delta})$ in Equation 8, our adapter reduces to several classic methods in literature. For example, if $f$ is linear and $g(\cdot)$ is $L^1$-norm for $\boldsymbol{\delta}$, denote by $\widetilde{y}_i = y_i - \boldsymbol{x}_i' \widehat{\boldsymbol{\theta}}_{\mathrm{s}}$. Then our objective is equivalent to the standard quantile Lasso model (Belloni & Chernozhukov, 2011), i.e.,

$$\widehat{\boldsymbol{\delta}} = \arg\min_{\boldsymbol{\delta} \in \mathbb{R}^d} \frac{1}{n} \sum_{i=0}^{n} \rho_\tau \left( \widetilde{y}_i - \boldsymbol{x}_i' \boldsymbol{\delta} \right) + \lambda \|\boldsymbol{\delta}\|_1 . \tag{9}$$

When the parameters of the model are matrices or tensors, $g(\cdot)$ should be set as the matrix nuclear norm to explicitly promote low-rank solutions.

Lastly, we comment that in our formulation, we add penalty/regularization as an extra term in objective instead of treating it as a separate constraint. It alleviates the challenge of training in many scenarios since equation Equation 8 is a unconstrained optimization and often convex (if $f(\boldsymbol{x}, \boldsymbol{\delta}), g(\boldsymbol{\delta})$ are convex). In practice, people can impose explicit constraints on $\delta$ in optimization as well, for example, ensuring a low-rank neural network structure on weight updates of format a multiplication of two low-dimensional matrices, i.e., the like LoRA-alike fine-tuning (Hu et al., 2022; Zhang et al., 2023; Wang & Liang, 2024). Those two types of formulation are closely connected.

## 3 THEORETICAL ANALYSIS: STATISTICAL GUARANTEES FOR LINEAR ADAPTER

In this section, we establish a theoretical analysis to our quantile regression adapter. We mainly focus on the high-dimensional setting where the sample size of target task is much less than the feature number. Otherwise, direct training is sufficient to recover good solutions and the benefits of transfer learning is marginal. To simplify, we restrict our discussions to high-dimensional linear model only. The reasons why we choose linear model as the object of study are twofold. First, statistical theory on linear models are well-developed, especially in the high-dimensional regime. Therefore, We can borrow the rich existing tools to analyze the behavior of transfer learning. Second, linear model is simple enough to clearly illustrate when and why quantile regression adapter can work. With appropriate tools, those insights can be generalized to nonlinear models like neural network as well.

Specifically, we assume that the conditional quantile model is linear, i.e., $f(\boldsymbol{x}; \boldsymbol{\theta}) = \boldsymbol{x}' \boldsymbol{\theta}$. In this case, the linear quantile regression can be expressed as $y = \boldsymbol{x}' \boldsymbol{\theta}^\star + \epsilon$, where $\epsilon$ denotes the noise in observation that satisfies the quantile condition $P(\epsilon \leq 0) = \tau$. We choose the vector $L^1$-norm as the regularization term. Then the objective in Equation 8 becomes

$$\mathcal{L}^a(\boldsymbol{\delta}; \mathcal{D}) = \frac{1}{n} \sum_{i=0}^{n} \rho_\tau \left( y_i - \boldsymbol{x}_i'(\widehat{\boldsymbol{\theta}}_{\mathrm{s}} + \boldsymbol{\delta}) \right) + \lambda \|\boldsymbol{\delta}\|_1 , \tag{10}$$

By setting $\boldsymbol{\theta} = \widehat{\boldsymbol{\theta}}_{\mathrm{s}} + \boldsymbol{\delta}$, we obtain

$$\widehat{\boldsymbol{\theta}} = \arg\min_{\boldsymbol{\theta} \in \mathbb{R}^d} \frac{1}{n} \sum_{i=0}^{n} \rho_\tau \left( y_i - \boldsymbol{x}_i' \boldsymbol{\theta} \right) + \lambda \left\| \boldsymbol{\theta} - \widehat{\boldsymbol{\theta}}_{\mathrm{s}} \right\|_1 , \tag{11}$$

which exhibits similar structure as the objective in quantile Lasso method but the center of deviation penalty becomes $\widehat{\boldsymbol{\theta}}_{\mathrm{s}}$, the estimated parameter of source task. Such an analogy motivates us to adapt the quantile Lasso theory to study the properties of linear quantile regression adapter. However,

since $\widehat{\boldsymbol{\theta}}_{\mathrm{s}}$ is not perfect, the estimation error with true parameter in source task $\boldsymbol{\theta}^{\star}$ may impact the performance task parameter estimation. Incorporating this error into the analysis of $\widehat{\boldsymbol{\theta}}$ is nontrivial.

Before we present our theoretical results, we first introduce some regularity conditions. We begin with an assumption about data distribution.

**Assumption 3.1** (Data Setting). *Each downstream data point in $\mathcal{D}$ is i.i.d. drawn from a distribution $(\boldsymbol{x}, y) \sim p$. For covariate $\boldsymbol{x}$, the conditional density $f(y|\boldsymbol{x})$ is continuously differentiable with uniform upper bounds $\bar{f}$ and $\bar{f}'$ for value $f(y|\boldsymbol{x})$ and derivative $\nabla_y f(y|\boldsymbol{x})$, respectively. Furthermore, there exists a positive constant $\underline{f}$ such that $f(y|\boldsymbol{x}) > \underline{f} > 0$ for all $y$ and $\boldsymbol{x}$. Furthermore, without loss of generality, we standardize $\boldsymbol{x}$ with zero mean and unit standard error.*

In next, we introduce some concepts and assumptions related to distributional shift. We first introduce a condition to quantify the transferability between target and source data.

**Definition 3.2** (Restricted Set and Restricted Eigenvalue Condition). *Let $\mathbb{S} = supp(\boldsymbol{\theta}) := \{j \subseteq [d] : |\boldsymbol{\theta}_j| > 0\}$ be the support of a fixed vector $\boldsymbol{\theta} \in \mathbb{R}^d$, we define $\mathbb{A}(\mathbb{S}, \alpha)$ the restricted set of parameter $\alpha$ as*

$$\mathbb{A}(\mathbb{S}, \alpha) = \{\boldsymbol{\delta} \in \mathbb{R}^d : \|\boldsymbol{\delta}_{\mathbb{S}^c}\|_1 \leq \alpha \|\boldsymbol{\delta}_{\mathbb{S}}\|_1, \alpha \geq 0\}.$$

*Moreover, we say the covariance matrix $\boldsymbol{\Sigma} \in \mathbb{R}^{d \times d}$ and index set $\mathbb{S} \subseteq [d]$ meet the Restricted Eigenvalue (RE) Condition for constant $\kappa > 0$ when*

$$\|\boldsymbol{\delta}_{\mathbb{S}}\|_1 \leq \frac{\sqrt{|\mathbb{S}|}}{\kappa} \left\|\boldsymbol{\Sigma}^{1/2}\boldsymbol{\delta}\right\|_2, \tag{12}$$

*for all $\boldsymbol{\delta} \in \mathbb{A}(\mathbb{S}, \alpha)$.*

The restricted eigenvalue (RE) condition is a standard assumption in the high-dimensional statistics literature in order to establish convergence rate for Lasso-type estimator in high-dimensional regime (Tibshirani, 1996; Bickel, 2007; Raskutti et al., 2010; Wainwright, 2019). In general, the identifiability of structural parameter of linear regression depends on the positive-definiteness of sample covariance matrix. In high-dimensional regime where the feature dimension is much larger than sample size, the sample covariance matrix in unlikely to be positive-definite for the whole parameter space. The RE condition relaxes this requirement to a smaller subspace $\mathbb{A}(\mathbb{S}, \alpha)$ instead. We refer to Wainwright (2019) for more discussions on the RE condition. In summary, we adopt the RE condition in this paper to ensure that the bias $\boldsymbol{\theta}^{\star} - \boldsymbol{\theta}_{\mathrm{s}}^{\star}$ is identifiable in the scenario of $d \gg n$. Additionally, if in the non-high-dimensional regime, i.e., $n \gg d$, the covariance matrix $\boldsymbol{\Sigma}$ is positive-definite and the RE condition is automatically satisfied (Raskutti et al., 2010; Wainwright, 2019).

Based on the RE condition, we impose the following assumption.

**Assumption 3.3** (Transferability Condition). *Let $\boldsymbol{\delta}^{\star} = \boldsymbol{\theta}^{\star} - \boldsymbol{\theta}_{\mathrm{s}}^{\star}$ be the difference of true parameters of target and source data. The restricted eigenvalue condition is satisfied for index set $\mathbb{S} = supp(\boldsymbol{\delta}^{\star})$ and target covariance matrix $\boldsymbol{\Sigma}$ with some positive constant $\kappa$. Furthermore, the sparsity coefficient $s = |\mathbb{S}|$ is much smaller than feature's dimension $d$ and target data sample size $n$.*

Assumption 3.3 uses the concept of sparsity to measure distributional shift and assumes that the difference in true parameters of target and source model is sparse. That is to say, in most dimensions, the parameters that determine target and source model are the same. It is an appropriate assumption in our setting since only when the true parameters are largely overlapped, transferring knowledge from pretrained model to donwstream target task is theoretically beneficial. In this case, the information stored in the parameters of the pretrained model can be directly applied to target task, which motivates the adapter-tuning strategy. We only need to use the extra target data to learn the low-dimensional discrepancy, which is achievable even if target dataset is limited like $n \ll d$. In what follows, we use the sparsity coefficient $s$ to denote the number of non-zero values in $\boldsymbol{\theta}^{\star} - \boldsymbol{\theta}_{\mathrm{s}}^{\star}$, i.e., $s = \|\boldsymbol{\delta}^{\star}\|_0$. The sparse coefficient $s$ determines the magnitude of distributional shift, as well as the intrinsic difficulty of transfer learning. In an extreme case where $s = 0$, i.e., the source and target models are exactly the same, applying the pretrained model to target task is trivially good. On the other hand, if $s$ is close to $d$, we should not expect transferring knowledge in pretrained model directly to target model, and thus, it is hard to learn ideally with limited extra data. As a result, our subsequent theoretical analysis mainly focuses on the nontrivial regime where $d \gg n \gg s$.

Lastly, we impose a regularity condition on the curvature of covariate $\boldsymbol{x}$'s distribution that ensures certain growth rate and non-degeneration.

**Assumption 3.4** (Bounded and Restricted Growth Condition). *There exists a constant $b \in \mathbb{R}$ such that $\|\boldsymbol{\theta}^\star\|_1 \leq b$. Additionally, we assume that for any target sample $\boldsymbol{x}_i \in \mathbb{R}^d$, and for any target estimator $\widehat{\boldsymbol{\theta}} \neq \boldsymbol{\theta}^\star$, the following holds:*

$$q := \frac{3}{8} \frac{\underline{f}^{3/2}}{\bar{f}'} \inf_{\tau \in (0,1)} \frac{\mathbb{E}[|\boldsymbol{x}_i'(\widehat{\boldsymbol{\theta}} - \boldsymbol{\theta}^\star)|^2]^{3/2}}{\mathbb{E}[|\boldsymbol{x}_i'(\widehat{\boldsymbol{\theta}} - \boldsymbol{\theta}^\star)|^3]} > 0.$$

In assumption 3.4, the upper bound of $b$ is used to characterize the worst-case parameter magnitude of $\boldsymbol{\theta}^\star$, which is standard. It also measures the relationship between the expected values of the squared and cubic powers of the residual. Assumption 3.4 is adapted from the statistical literature on quantile Lasso method (Belloni & Chernozhukov, 2011), which builds the foundation of our analysis. We will apply restricted growth condition to control the minoration of the quantile regression objective function by a quadratic function in our proof.

In our setting, since the true parameter of source model is unknown and estimated via the pretrained model, it is necessary to consider the impact of such an estimation error on the performance of target task. Let $\boldsymbol{\nu} = \widehat{\boldsymbol{\theta}}_{\mathrm{s}} - \boldsymbol{\theta}_{\mathrm{s}}^\star \in \mathbb{R}^d$ be the estimation error of $\widehat{\boldsymbol{\theta}}_{\mathrm{s}}$. Then if the source dataset is sufficient and the pretrained model is correctly specified, we expect $\boldsymbol{\nu}$ should be small. For example, if the source task is a linear quantile/lease-square regression with sample size $n_s \gg d$, under mild conditions, it holds that $\|\boldsymbol{\nu}\|_2 = \mathcal{O}((d/n_s)^{1/2})$. Nevertheless, the presence of a non-zero $\boldsymbol{\nu}$ prevents us quoting the exisitng results of quantile Lasso directly. We have to carefully tailor the quantile Lasso analysis framework in order to accommodate the interplay of estimation errors in two tasks. With above preparations, we are ready to present our main theoretical results.

**Theorem 3.5** (Convergence Rate of Linear Quantile Regression Adapter). *Let $\widehat{\boldsymbol{\theta}}$ be the optimal solution to optimization Equation 11 and the regularization hyperparameter $\lambda$ is set as*

$$\lambda^\star \asymp \max\left\{ \sqrt{\frac{\log(d) + u}{n}}, \; d\,\|\boldsymbol{\nu}\|_2 \right\}. \tag{13}$$

*Under assumptions 3.1, 3.3, and 3.4, with probability at least $1 - \exp(-u)$ for some $u > 0$, the estimation error of our linear quantile regression adapter is upper bounded as*

$$\left\|\widehat{\boldsymbol{\theta}} - \boldsymbol{\theta}^\star\right\|_1 \leq \mathcal{O}\left( \max\left\{ s\sqrt{\frac{\log(d) + u}{n}}, \; ds\,\|\boldsymbol{\nu}\|_2 \right\} \right). \tag{14}$$

Theorem 3.5 establishes the convergence rate of the target task estimation error. To highlight the insights, we only present the impact of factors $s, d, n, \nu$ in Theorem 3.5 and omit other constant factors which are problem-specific. Note that our error bound is the maximum of two terms. The first term primarily depends on sparsity parameter $s$ *linearly* and decays to zero at rate of $n^{-1/2}$. The dependency on $d$ is logarithmic. The second term is inherited from the source task estimation error $\boldsymbol{\nu}$. If the source dataset sample size $n_{\mathrm{s}}$ is sufficiently large and the pretrained model fits well, then $\|\boldsymbol{\nu}\|_2$ is of order $\mathcal{O}(n_s^{-1/2})$ and thus, negligible. In this case, the first term dominates.

As contrast, if we do not use transfer learning or pretrained model adaptation, and rely on target data only to train the quantile regression model, the convergence rate for estimation error is expected to depend on feature's dimension $d$ linearly rather than $s$, which is trivial in high-dimensional regime. Such a comparison shows the power of our quantile regression adapter. Additionally, Theorem 3.5 also requires an appropriate magnitude of the regularization hyperparameter $\lambda$ in order to ensure the desired convergence rate. Intuitively, if $\lambda$ is too large, the target estimator may fail to learn new knowledge from the target data. Similarly, $\lambda$ cannot be too small, as the target model needs to retain and leverage the general representations learned from the pretrained model. Our insights largely match the results in classic quantile Lasso theory as well (Belloni & Chernozhukov, 2011).

As a corollary of Theorem 3.5, we can also establish the bound for the prediction error in target task. Specifically, consider a clipped target estimator defined as $\widehat{\boldsymbol{\theta}}^{\mathrm{CLIP}} = \widehat{\boldsymbol{\theta}}$ if $\|\widehat{\boldsymbol{\theta}}\|_1 \leq 2b$ where $2b$ is the maximal possible $L^1$ norm of the true parameter, and $\widehat{\boldsymbol{\theta}}^{\mathrm{CLIP}} = 0$ otherwise. Similarly, setting the

tuning parameter $\lambda$ as

$$\lambda^{\star} \asymp \max \left\{ \sqrt{\frac{\log(bdn)}{n}}, \, d \left\| \boldsymbol{\nu} \right\|_2 \right\}. \tag{15}$$

Then the expected out-of-sample prediction error for any new input $\boldsymbol{x}$ can be upper bounded

$$\mathbb{E}\left[\left\| \boldsymbol{x}'\widehat{\boldsymbol{\theta}}^{\text{CLIP}} - \boldsymbol{x}'\boldsymbol{\theta}^{\star} \right\|_1\right] = \mathcal{O}\left( \max\left\{ \frac{s \left\| \boldsymbol{x} \right\|_{\infty}}{\sqrt{n}} \log(dn), \, ds \left\| \boldsymbol{\nu} \right\|_2 \left\| \boldsymbol{x} \right\|_{\infty} \right\} \right). \tag{16}$$

## 4 EXPERIMENT

In this section, we conduct numerical experiments to demonstrate the performance of our quantile regression adapter and verify theoretical results. We aim to answer the following questions: (1) Under what conditions is quantile adaptation efficient for new downstream task? (2) Does quantile adapter perform better than Lasso-style adapter?

**Data setting:** We perform a simulation study with sample sizes $n_s = 1000$ for the source data, $n = 150$ for the target data and $n_{\text{eval}} = 1000$ for the evaluation data. The $n_s$ source observations, denoted as $\boldsymbol{x}$, are drawn from a $d$-dimensional multivariate standard normal distribution with $d = 100$. The first $n$ samples of the source data observations are used as target data observations, and the evaluation data observations are generated independently in the same way. The true parameter for the source domain is fixed as $\boldsymbol{\theta}_{\text{s}}^{\star} = \{1, \ldots, 1\}' \in \mathbb{R}^d$. To obtain the target model $\boldsymbol{\theta}^{\star}$, we generate $\delta^{\star}$ by uniformly setting $s$ elements to 0.9 and the remaining elements to 0. The responses are generated as $y_s = \boldsymbol{x}_s' \boldsymbol{\theta}_s^{\star} + \epsilon_s$ for the source pairs $(\boldsymbol{x}_s, y_s) \sim \mathcal{D}_s$, $y = \boldsymbol{x}'\boldsymbol{\theta}^{\star} + \epsilon$ for the target pairs $(\boldsymbol{x}, y) \sim \mathcal{D}$, and $y_{\text{eval}} = \boldsymbol{x}_{\text{eval}}' \boldsymbol{\theta}^{\star} + \epsilon_{\text{eval}}$ for the evaluation data. All noise terms are i.i.d. from the standard normal distribution $\mathcal{N}(0, 1)$.

**Under what conditions is quantile adaptation efficient for new downstream task?** To assess the efficiency of our quantile adaptation method, we first instantiate the pretrained model by optimizing sample mean squared error (MSE). We then train our quantile estimator to predict the median of the responses using Equation 10, with quantile level $\tau = 0.5$, and evaluate it with $\widehat{\boldsymbol{\delta}} + \widehat{\boldsymbol{\theta}}_{\text{s}}$. We refer to our adapter as **QAdapter**. We compare our method against three baselines: (1) **Direct Training (DT)** that directly optimizes the linear quantile estimator without utilizing the pretrained model; (2) **Zero-shot** that directly evaluates the performance of the pretrained model on the test data without any adaption. (3) **Average** that combines the parameters of the pretrained model and DT by $\alpha_1 \widehat{\boldsymbol{\theta}}_s + (1 - \alpha_1)\widehat{\boldsymbol{\theta}}$, where $\alpha_1 \in (0, 1)$. Following previous works as in (Bastani, 2021; Jin et al., 2023; Li et al., 2022), we use MSE to evaluate the estimation performance of the downstream models, i.e., $\|\widehat{\boldsymbol{\theta}} - \boldsymbol{\theta}^{\star}\|_2$.

Figure 1a illustrates the performance of estimation task under different values of the similarity coefficient $s$. Our results show that QAdapter achieves state-of-the-art performance when estimating the true target model in downstream tasks. We attribute the failure of Zero-shot estimation to the discrepancy between the source and target true models. Meanwhile, DT performs poorly when target data are scarce, as it fails to utilize the knowledge of the pretrained model. We also note that, when the source model is equal to the target model ($s = 0$), the performance of QAdapter is close to that of Zero-shot. However, as $s$ increases to 100, QAdapter's performance deteriorates to that of DT, suggesting that the pretrained model becomes less useful for the downstream task. In addition, the estimation error of the QAdapter increases linearly as $s$ increases. These observations are consistent with our Theorem 3.5.

Figures 1b and 1c depicts the performance of the estimation task under different values of $\lambda$ and $n$, respectively. Specifically, we fix the sparsity coefficient at $|\mathbb{S}| = 20$ and, iterate over $\lambda$ and $n$ with fixed step sizes respectively to show the trend of estimation error. As the Figure 1b shown, the choice of $\lambda$ can substantially affect the adaptation performance. Figure 1c show that the estimation error seemingly degrade at the rate of $\mathcal{O}(n^{-1/2})$ and our transfer learning algorithm has significant benefit when target sample is small. More details of the above implementation and prediction results can be found in the Appendix B.

**Does quantile adapter perform better than Lasso-style adapter?** For the second part of our numerical experiments, we compare QAdapter with another Lasso objective as appeared in Bastani

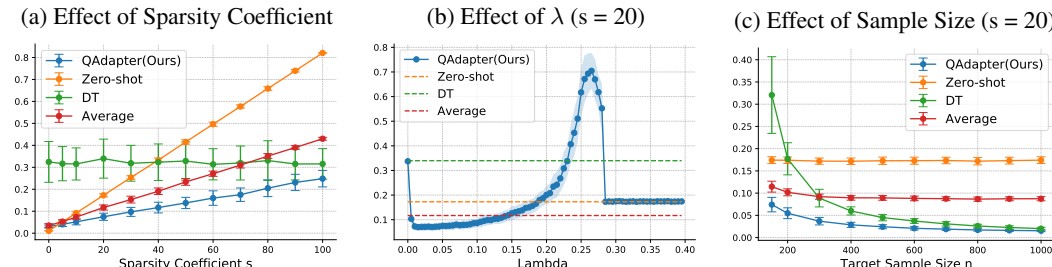

Figure 1: Analysis of various factors affecting estimation error of model, measured using $\|\widehat{\boldsymbol{\theta}} - \boldsymbol{\theta}^\star\|_2$ on the y-axis. (a) The effect of the sparsity coefficient $s$. Our QAdapter method consistently achieves lower estimation errors compared to other methods. (b) The effect of $\lambda$. Using excessively high and low values of $\lambda$ can degrade performance. (c) The effect of target data $n$. The lower the amount of data for downstream tasks, the greater the necessity of using the quantile adapter.

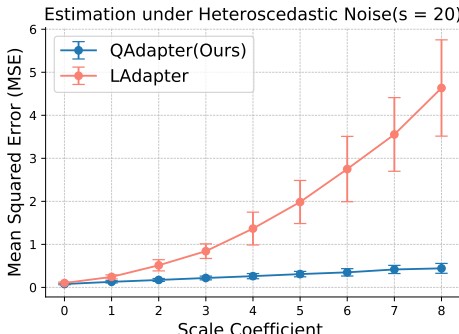

Figure 2: We evaluate the downstream estimation error of different adaptation methods under heteroscedastic downstream tasks. The target sample is generated as $y_i = \boldsymbol{x}_i'\boldsymbol{\theta}^\star + \mathcal{N}(0,1) \times (1 + \text{scale} \times \boldsymbol{x}_{i1})$. As the scale coefficient increases, the extent of disturbance from heteroscedastic noise is enhanced, causing LAdapter to collapse. On the other hand, the performance of QAdapter ($\tau = 0.5$) exhibits lower disturbance.

(2021) and Li et al. (2022), where the task-specific parameters are trained by

$$\text{LAdapter:} \quad \widehat{\boldsymbol{\delta}}_L = \underset{\boldsymbol{\delta} \in \mathbb{R}^d}{\arg\min} \, \frac{1}{n} \sum_{i=0}^{n} \left( y - \boldsymbol{x}'(\widehat{\boldsymbol{\theta}}_s + \boldsymbol{\delta}) \right)^2 + \|\boldsymbol{\delta}\|_1 . \tag{17}$$

We refer to this method as **LAdapter**. To demonstrate the robustness of adapting pretrained models to heteroscedastic data, where the variance of the noise is not consistent across all data points, we generate the target data by $y_i = \boldsymbol{x}_i'\boldsymbol{\theta}^\star + \mathcal{N}(0,1) \times (1 + \text{scale} \times \boldsymbol{x}_{i1})$ for $i = 1, \dots, n$ with $s = 20$, and keep other settings unchanged. Figure 2 compares the estimation performance across different values of the scale coefficient. The results show that LAdapter struggles to capture the true model information when subjected to heteroscedastic noise.

Additionally, we consider the downstream task of extreme value prediction. We generate target data by randomly assigning 10% of the samples to follow $y_i = \boldsymbol{x}_i'\boldsymbol{\theta}^\star + \mathcal{N}(0,1)$ while keeping the remaining 90% as $y_i = 0 + \mathcal{N}(0,1)$. In this case, the 90% of the data provides no information about the model coefficients, and the 10% represents rare, worst-case events that are highly informative yet costly. We evaluate the accuracy of the adapters in estimating the true parameters, as shown in Table 1. Our results indicate that LAdapter fails to learn the true model due to the scarcity of informative data; in contrast, the quantile adapter with $\tau = 0.9$ performs significantly better, as its design allows it to capture this portion of the distribution more effectively.

Table 1: Comparison of Adaptation Methods in Extreme Value Prediction (s = 20)

|  | QAdapter ($\tau = 0.9$) | QAdapter ($\tau = 0.5$) | LAdapter |
|---|---|---|---|
| $\|\widehat{\boldsymbol{\theta}} - \boldsymbol{\theta}^\star\|_2$ | **0.18 ± 0.01** | 2.75 ± 0.24 | 36.77 ± 6.40 |
| Quantile Loss | **3.93 ± 0.06** | 4.90 ± 0.40 | 23.66 ± 2.29 |

## 5 CONCLUSION

In this work, we propose an efficient quantile regression algorithm via transfer learning, specifically designed to transfer knowledge to risk-sensitive downstream tasks. We introduce a measure to theoretically quantify the transferability of knowledge and provide statistical guarantees for adaptation efficiency under a linear structural model. An interesting direction for future research could involve relaxing the linear form assumption and extending the method to more general adaptation functions. We also believe that developing practical implementations of quantile transfer learning methods for real-world downstream tasks can be an important direction for future work.

## ETHICS STATEMENT

We have adhered to the ethical standards and practices as suggested in the ICLR Code of Ethics. Our study does not involve human subjects and not publicly available datasets are employed. We have taken care to ensure that our quantile regression algorithm is designed to minimize biases and promote fairness, recognizing the potential implications of its application in risk-sensitive domains. By providing statistical guarantees and measures of transferability, we aim to enhance the reliability and ethical deployment of our methods. All aspects of our research have been carried out with integrity, maintaining transparency and reproducibility to support the responsible advancement of knowledge in this field.

## REPRODUCIBILITY STATEMENT

we provide clear explanations of after impose assumptions in paper, and a complete proof of our theorem can be found in Appendix A. Additionally, all experiments and results reported in this paper can be reproduced using the provided anonymous source code at `https://anonymous.4open.science/r/QAdapter-5FF6`. We discuss the all detailed code implementation in Appendix B.

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

## A  PROOF OF THEOREM 3.5 AND COROLLARY

In this section, we will show the detailed proof for the parameter estimation error of the linear estimator in Equation 10. In Subsection A.1, we introduce several additional pieces of useful notation and formulation throughout the section for convenience. Secondly, we establish some technical lemmas for our proof in Subsection A.2. Lastly, we provide the completed proof in Subsection A.3.

### A.1  QUANTILE TRANSFER LEARNING IN LINEAR CASES

In Section 3, we propose the statistics results for our transfer learning framework under linear target estimator. Recall that we assume that the response of the downstream task can be formulated as a linear function, that is

$$\begin{cases} y = \boldsymbol{x}'\boldsymbol{\theta}^\star + \epsilon \\ \mathrm{P}\left(\epsilon \leq 0\right) = \tau, \end{cases} \qquad \forall(\boldsymbol{x}, y) \sim p. \tag{18}$$

In that case, we use the linear approximation function $f(\boldsymbol{x}, \boldsymbol{\theta} + \boldsymbol{\delta}) = \boldsymbol{x}'(\boldsymbol{\theta} + \boldsymbol{\delta})$ to estimate the true coefficient of target model, our optimization objective in downstream task can be written as

$$\frac{1}{n}\sum_{i=0}^{n}\rho_\tau\left(y_i - \boldsymbol{x}'_i(\widehat{\boldsymbol{\theta}}_{\mathrm{s}} + \boldsymbol{\delta})\right) + \lambda\|\boldsymbol{\delta}\|_1, \tag{19}$$

where $\rho_\tau(x) = x(\tau - \mathbf{1}_{x\leq 0})$ is the standard quantile loss function with quantile level $\tau \in (0,1)$. Note that the only trainable parameter in adaptation stage is $\boldsymbol{\delta}$. We define the accumulated empirical quantile loss in $\mathcal{D}$ as

$$\widehat{\mathcal{R}}_\tau(\boldsymbol{\delta}) := \frac{1}{n}\sum_{i=0}^{n}\rho_\tau(y_i - \boldsymbol{x}'_i(\widehat{\boldsymbol{\theta}}_{\mathrm{s}} + \boldsymbol{\delta})),$$

and $\mathcal{R}_\tau(\boldsymbol{\delta}) := \mathbb{E}_{(x,y)\sim p}\widehat{\mathcal{R}}_\tau$. The our estimator is then simply $\widehat{\boldsymbol{\theta}} = \widehat{\boldsymbol{\theta}}_{\mathrm{s}} + \widehat{\boldsymbol{\delta}}$, where $\widehat{\boldsymbol{\delta}}$ is estimated in Equation 19. Similarly, the true target estimator is $\boldsymbol{\theta}^\star = \widehat{\boldsymbol{\theta}}_{\mathrm{s}} + \widetilde{\boldsymbol{\delta}}$, where $\widetilde{\boldsymbol{\delta}}$ can be obtained in the following objective

$$\widetilde{\boldsymbol{\delta}} = \arg\min_{\boldsymbol{\delta}\in\mathbb{R}^d} \mathcal{R}_\tau(\boldsymbol{\delta}) + \lambda\|\boldsymbol{\delta}\|_1. \tag{20}$$

We will alternately use these two notations in our proof, which are unambiguous and equivalent:

$$\left\|\widehat{\boldsymbol{\delta}} - \widetilde{\boldsymbol{\delta}}\right\|_1 = \left\|\widehat{\boldsymbol{\theta}} - \widehat{\boldsymbol{\theta}}_{\mathrm{s}} - \boldsymbol{\theta}^\star + \widehat{\boldsymbol{\theta}}_{\mathrm{s}}\right\|_1 = \left\|\widehat{\boldsymbol{\theta}} - \boldsymbol{\theta}^\star\right\|_1. \tag{21}$$

Moreover, we denote the following event $\mathcal{J}_{(\boldsymbol{\delta},\boldsymbol{\delta}')}$ by

$$\mathcal{J}_{(\boldsymbol{\delta},\boldsymbol{\delta}')} := \left\{\sup_{\|\boldsymbol{\delta}-\boldsymbol{\delta}'\|_1\leq t}\left\|\widehat{\mathcal{R}}_\tau(\boldsymbol{\delta}) - \widehat{\mathcal{R}}_\tau(\boldsymbol{\delta}') - (\mathcal{R}_\tau(\boldsymbol{\delta}) - \mathcal{R}_\tau(\boldsymbol{\delta}'))\right\|_1 \leq \lambda_0 t\right\},$$

where $t$ and $\lambda_0$ are some positive scalar. We define the complement of the event as:

$$\mathcal{J}^C_{(\boldsymbol{\delta},\boldsymbol{\delta}')} := \left\{\sup_{\|\boldsymbol{\delta}-\boldsymbol{\delta}'\|_1\leq t}\left\|\widehat{\mathcal{R}}_\tau(\boldsymbol{\delta}) - \widehat{\mathcal{R}}_\tau(\boldsymbol{\delta}') - (\mathcal{R}_\tau(\boldsymbol{\delta}) - \mathcal{R}_\tau(\boldsymbol{\delta}'))\right\|_1 > \lambda_0 t\right\}.$$

### A.2  TECHNICAL LEMMAS FOR THEOREM 3.5

We next establish several useful lemmas for our proof.

**Lemma A.1** (Lipschitz Continuity). *For any vector $\boldsymbol{x} \in \mathbb{R}^d$, scalar $y \in \mathbb{R}$ and quantile level $\tau \in (0,1)$, the quantile loss function $\rho_\tau(y - \boldsymbol{x}'\boldsymbol{\theta})$ is Lipschitz continuous with a Lipschitz constant $L_\tau > 0$ that depends on $\tau$. Specifically, for different two parameters $\boldsymbol{\theta}_1, \boldsymbol{\theta}_2 \in \mathbb{R}^d$, we have*

$$\|\rho_\tau(y - \boldsymbol{x}'\boldsymbol{\theta}_1) - \rho_\tau(y - \boldsymbol{x}'\boldsymbol{\theta}_2)\|_1 \leq L_\tau \|\boldsymbol{x}'(\boldsymbol{\theta}_1 - \boldsymbol{\theta}_2)\|_1.$$

The proof is completed by a categorical discussion of the intervals of the quantile loss function.

**Lemma A.2** (Control the empirical error of $\widehat{\mathcal{R}}_\tau$). *With* $\lambda_0 \geq \sqrt{8L_\tau^2/n}$, *we have,*

$$P\left(\mathcal{J}_{(\widehat{\boldsymbol{\delta}},\widetilde{\boldsymbol{\delta}})}\right) \geq 1 - 8d \cdot \exp\left(-\frac{\lambda_0^2 \cdot n \cdot \kappa^2}{32L_\tau^2}\right).$$

*Proof of Lemma A.2.* To simplify notation, we denote:

$$\Delta := \left\|\widehat{\boldsymbol{\delta}} - \widetilde{\boldsymbol{\delta}}\right\|_1.$$

The proof is mainly based on the symmetrization lemma for probabilities. Using the Corollary 3.4 in (Geer, 2000), we have for $\lambda_0 \geq \sqrt{8L_\tau^2/n}$,

$$\mathrm{P}\left(\mathcal{J}_{(\widehat{\boldsymbol{\delta}},\widetilde{\boldsymbol{\delta}})}^C\right)$$

$$\leq 4\mathrm{P}\left(\sup_{\Delta \leq t}\left\|\frac{1}{n}\sum_{i=1}^n W_i \cdot \left(\rho_\tau\left(y_i - \boldsymbol{x}_i'(\widehat{\boldsymbol{\delta}} + \widehat{\boldsymbol{\theta}}_{\mathrm{s}})\right) - \rho_\tau\left(y_i - \boldsymbol{x}_i'(\widetilde{\boldsymbol{\delta}} + \widehat{\boldsymbol{\theta}}_{\mathrm{s}})\right)\right)\right\|_1 > \frac{\lambda_0 t}{4}\right) \quad (22)$$

$$\leq 4\mathrm{P}\left(\sup_{\Delta \leq t}\left\|\frac{L_\tau}{n}\sum_{i=1}^n W_i \cdot \boldsymbol{x}_i'\left(\widehat{\boldsymbol{\theta}} - \boldsymbol{\theta}^\star\right)\right\|_1 > \frac{\lambda_0 t}{4}\right),$$

where $(W_1, ..., W_n)$ is the Rademacher sequence independent of samples $\mathcal{D}$, and i.i.d. with probability $\mathrm{P}(W_i = 1) = \mathrm{P}(W_i = -1) = \frac{1}{2}$, and the last inequality holds by Lemma A.1. Moreover, by the Cauchy–Schwarz inequality, we have for any vector $\boldsymbol{a}, \boldsymbol{b} \in \mathbb{R}^d$, $\|\boldsymbol{ab}\|_1 = \|\boldsymbol{a}\|_1 \|\boldsymbol{b}\|_\infty$. With $\boldsymbol{a} = \boldsymbol{\theta}^\star - \widehat{\boldsymbol{\theta}}$ and $\boldsymbol{b} = \sum_{i=1}^n W_i \boldsymbol{x}_i$, we can obtain the following inequality

$$\left\|\sum_{i=1}^n W_i \cdot \boldsymbol{x}_i'(\widehat{\boldsymbol{\theta}} - \boldsymbol{\theta}^\star)\right\|_1 \leq \left\|\widehat{\boldsymbol{\theta}} - \boldsymbol{\theta}^\star\right\|_1 \max_{j \leq d}\left\|\sum_{i=1}^n W_i \boldsymbol{x}_{ij}\right\|_1. \quad (23)$$

where, $d$ is the dimension of vector $\boldsymbol{x}$, and $\boldsymbol{x}_{ij}$ denotes the $j$th component of vector $\boldsymbol{x}_i$. Hence, applying the markov inequality to further bound the right-hand side of Equation 22, we have for any $\xi > 0$,

$$4\mathrm{P}\left(\sup_{\Delta \leq t}\left\|\frac{L_\tau}{n}\sum_{i=1}^n W_i \cdot \boldsymbol{x}_i'(\widehat{\boldsymbol{\theta}} - \boldsymbol{\theta}^\star)\right\|_1 > \frac{\lambda_0 t}{4}\right)$$

$$\leq \min_{\xi > 0} 4\exp\frac{-\xi\lambda_0 t}{4} \cdot \mathbb{E}\left[\exp\left(\frac{\xi L_\tau}{n}\sup_{\Delta \leq t}\left\|\sum_{i=1}^n W_i \cdot \boldsymbol{x}_i'(\widehat{\boldsymbol{\theta}} - \boldsymbol{\theta}^\star)\right\|_1\right)\right] \quad (24)$$

$$\leq \min_{\xi > 0} 4\exp\frac{-\xi\lambda_0 t}{4} \cdot \mathbb{E}\left[\exp\left(\frac{\xi L_\tau}{n}\sup_{\Delta \leq t}\left[\left\|\widehat{\boldsymbol{\theta}} - \boldsymbol{\theta}^\star\right\|_1 \max_{j \leq d}\left\|\sum_{i=1}^n W_i \boldsymbol{x}_{ij}\right\|_1\right]\right)\right],$$

where the last inequality holds by Equation 23. We obtain

$$\sup_{\Delta \leq t}\left[\left\|\widehat{\boldsymbol{\theta}} - \boldsymbol{\theta}^\star\right\|_1 \max_{j \leq d}\left\|\sum_{i=1}^n W_i \boldsymbol{x}_{ij}\right\|_1\right] \leq \sup_{\Delta \leq t}\left[\Delta \cdot \max_{j \leq d}\left\|\sum_{i=1}^n W_i \boldsymbol{x}_{ij}\right\|_1\right]$$

$$= t \cdot \max_{j \leq d}\left\|\sum_{i=1}^n W_i \boldsymbol{x}_{ij}\right\|_1, \quad (25)$$

where the supremum is eliminated since the maximum value is attained when $\Delta = t$. Moreover, note that with the exchange rule of the expectation and maximum, we have such inequality:

$$\mathbb{E}\left[\max_{j \leq d}\exp\left\|\sum_{i=1}^n W_i \boldsymbol{x}_{ij}\right\|_1\right] \leq d\max_{j \leq d}\mathbb{E}\left[\exp\left\|\sum_{i=1}^n W_i \boldsymbol{x}_{ij}\right\|_1\right]. \quad (26)$$

Therefore, we combine Equation 25 and Equation 26, we then proceed to bound the right hand side of Equation 24, that is

$$\min_{\xi > 0} 4 \exp \frac{-\xi \lambda_0 t}{4} \cdot \mathbb{E} \left[ \exp \left( \frac{\xi L_\tau}{n} \cdot \sup_{\Delta \leq t} \left[ \left\| \boldsymbol{\theta}^\star - \widehat{\boldsymbol{\theta}} \right\|_1 \cdot \max_{j \leq d} \left\| \sum_{i=1}^n W_i \boldsymbol{x}_{ij} \right\|_1 \right] \right) \right] \tag{27}$$

$$\leq \min_{\xi > 0} 4 \exp \frac{-\xi \lambda_0 t}{4} \cdot \mathbb{E} \left[ \exp \left( \frac{\xi L_\tau}{n} \cdot t \cdot \max_{j \leq d} \left\| \sum_{i=1}^n W_i \boldsymbol{x}_{ij} \right\|_1 \right) \right] \tag{28}$$

$$\leq \min_{\xi > 0} \max_{j \leq d} 4d \cdot \exp \frac{-\xi \lambda_0 t}{4} \cdot \mathbb{E} \left[ \exp \left( \frac{\xi L_\tau}{n} \cdot t \cdot \left\| \sum_{i=1}^n W_i \boldsymbol{x}_{ij} \right\|_1 \right) \right], \tag{29}$$

where the first and second inequality holds by applying Equation 25 and Equation 26. Next, to eliminate the expectation, we adapt from the intermediate proof of the Hoeffding inequality, for self-contained purposes, we next show detailed derivation. For any scalar $a > 0$, and any column $0 \leq j \leq d$, we have

$$\mathbb{E} \left[ \exp \left( a \cdot \left\| \sum_{i=1}^n W_i \boldsymbol{x}_{ij} \right\|_1 \right) \right] = \mathbb{E} \left[ \mathbb{E} \left[ \exp \left( a \cdot \left\| \sum_{i=1}^n W_i \boldsymbol{x}_{ij} \right\|_1 \right) \middle| \boldsymbol{x}_{ij} \right] \right] \tag{30}$$

$$\leq \mathbb{E} \left[ \mathbb{E} \left[ \exp \left( a \cdot \sum_{i=1}^n W_i \boldsymbol{x}_{ij} \right) + \exp \left( -a \cdot \sum_{i=1}^n W_i \boldsymbol{x}_{ij} \right) \middle| \boldsymbol{x}_{ij} \right] \right] \tag{31}$$

$$= \prod_{i=1}^n \mathbb{E} \left[ \mathbb{E} \left[ \exp \left( a \cdot W_i \boldsymbol{x}_{ij} \right) + \exp \left( -a \cdot W_i \boldsymbol{x}_{ij} \right) \middle| \boldsymbol{x}_{ij} \right] \right] \tag{32}$$

$$= \prod_{i=1}^n \mathbb{E} \left[ \exp \left( a \cdot \boldsymbol{x}_{ij} \right) + \exp \left( -a \cdot \boldsymbol{x}_{ij} \right) \right] \tag{33}$$

$$\leq 2 \mathbb{E} \exp \left( \sum_{i=1}^n \frac{a^2 \boldsymbol{x}_{ij}^2}{2} \right) \tag{34}$$

$$= 2 \exp \left( a^2 \cdot \frac{n}{2} \right), \tag{35}$$

where the first equality holds by the law of iterated expectation, the next inequality by extension the absolute value and the monotone increase of exponential function, and the third and fourth equality holds by the property of the Rademacher sequence $(W_1, .... W_n)$, the last inequality holds by comparing Taylor's expansions of both sides. Last equality holds since we standardize the feature for each column $j$. Hence, applying the above result, we can simplify the right-hand side of Equation 27, that is

$$\min_{\xi > 0} \max_{j \leq d} 4d \cdot \exp \frac{-\xi \lambda_0 t}{4} \cdot \mathbb{E} \left[ \exp \left( \frac{\xi L_\tau}{n} \cdot t \cdot \left\| \sum_{i=1}^n W_i \boldsymbol{x}_{ij} \right\|_1 \right) \right]$$

$$\leq \min_{\xi > 0} 8d \cdot \exp \frac{-\xi \lambda_0 t}{4} \cdot \exp \left( \left( \frac{\xi L_\tau t}{n} \right)^2 \cdot \frac{n}{2} \right)$$

$$= \min_{\xi > 0} 8d \cdot \exp \left( \left( \frac{L_\tau t}{\sqrt{2n}} \right)^2 \xi^2 - \frac{\lambda_0 t}{4} \xi \right) \tag{36}$$

$$= 8d \cdot \exp \left( -\frac{\lambda_0^2 \cdot n}{32 L_\tau^2} \right),$$

where the last equality holds by optimizing objective $\exp(a\xi^2 + b\xi)$ with $\xi = -b/2a$. Combining Equation 22 and Equation 24 and Equation 27, and Equation 36, we can obtain

$$\mathrm{P} \left( \mathcal{J}_{(\widehat{\boldsymbol{\delta}}, \widetilde{\boldsymbol{\delta}})}^C \right) \leq 8d \cdot \exp \left( -\frac{\lambda_0^2 \cdot n}{32 L_\tau^2} \right). \tag{37}$$

Therefore, we have the event $\mathcal{J}_{(\widehat{\boldsymbol{\delta}}, \widetilde{\boldsymbol{\delta}})}$ holds with a high probability, i.e.

$$
\begin{aligned}
\mathrm{P}\left(\mathcal{J}_{(\widehat{\boldsymbol{\delta}}, \widetilde{\boldsymbol{\delta}})}\right) &= 1 - \mathrm{P}\left(\mathcal{J}^C_{(\widehat{\boldsymbol{\delta}}, \widetilde{\boldsymbol{\delta}})}\right) \\
&\geq 1 - 8d \cdot \exp\left(-\frac{\lambda_0^2 \cdot n}{32 L_\tau^2}\right).
\end{aligned}
\tag{38}
$$

$\square$

**Lemma A.3.** *On the event $\mathcal{J}_{(\widehat{\boldsymbol{\delta}}, \widetilde{\boldsymbol{\delta}})}$, and if assumption 3.1, 3.3, 3.4 holds, we have with $\lambda \geq 2\lambda_0 \geq \sqrt{8L_\tau^2/n}$*

$$
\left\|\widehat{\boldsymbol{\theta}} - \boldsymbol{\theta}^\star\right\|_1 \leq \frac{\underline{f}d}{\lambda}\|\boldsymbol{\nu}\|_2^2 + 8\|\boldsymbol{\nu}\|_1 + \frac{8\lambda s}{\underline{f}\kappa^2}.
$$

*provided $s$ obeys the growth condition*

$$
4q \geq \frac{\underline{f}d^{\frac{3}{2}}}{\lambda}\|\boldsymbol{\nu}\|_2^2 + 8\sqrt{d}\|\boldsymbol{\nu}\|_1 + \frac{8\lambda s\sqrt{d}}{\underline{f}\kappa^2}.
\tag{39}
$$

*Proof of Lemma A.3.* Proof by contradiction method. To simplify notation, let

$$
\Delta := \left\|\widehat{\boldsymbol{\delta}} - \widetilde{\boldsymbol{\delta}}\right\|_1, \quad t := \frac{\underline{f}d}{\lambda}\|\boldsymbol{\nu}\|_2^2 + 8\|\boldsymbol{\nu}\|_1 + \frac{8\lambda s}{\underline{f}\kappa^2}
$$

Recall that the $\widehat{\boldsymbol{\delta}}$ is any solution of the optimization problem in Equation 19. Given the Event $\mathcal{J}_{(\widehat{\boldsymbol{\delta}}, \widetilde{\boldsymbol{\delta}})}$ and assumption 3.3, we want show the event that

$$
\min_{\Delta \geq t} \widehat{\mathcal{R}}_\tau(\widehat{\boldsymbol{\delta}}) - \widehat{\mathcal{R}}_\tau(\widetilde{\boldsymbol{\delta}}) + \lambda\left\|\widehat{\boldsymbol{\delta}}\right\|_1 - \lambda\left\|\widetilde{\boldsymbol{\delta}}\right\|_1 < 0
\tag{40}
$$

is impossible, which suffices to prove the bound. Furthermore, we know that the objective function $\widehat{\mathcal{R}}_\tau$ is convex, and the left-hand side of the inequality in Equation 40 is convex. Hence we can replace $\Delta \geq t$ with $\Delta = t$ in Equation 40 while preserving the validity of our proof:

$$
\min_{\Delta = t} \widehat{\mathcal{R}}_\tau(\widehat{\boldsymbol{\delta}}) - \widehat{\mathcal{R}}_\tau(\widetilde{\boldsymbol{\delta}}) + \lambda\left\|\widehat{\boldsymbol{\delta}}\right\|_1 - \lambda\left\|\widetilde{\boldsymbol{\delta}}\right\|_1 < 0.
\tag{41}
$$

To ultimately invoke the transferability measure assumption 3.3, we need to express $\widehat{\boldsymbol{\delta}}$ in terms of its components in the index set. Recall that the notation of the bias term is denoted as $\boldsymbol{\delta}^\star := \boldsymbol{\theta}^\star - \boldsymbol{\theta}_{\mathrm{s}}^\star \in \mathbb{R}^d$. By definition and the triangle inequality, we have such a relationship.

$$
\begin{aligned}
\left\|\widehat{\boldsymbol{\delta}}\right\|_1 &= \left\|\widehat{\boldsymbol{\delta}}_{\mathbb{S}}\right\|_1 + \left\|\widehat{\boldsymbol{\delta}}_{\mathbb{S}^c}\right\|_1 \\
&\geq \|\boldsymbol{\delta}_{\mathbb{S}}^\star\|_1 - \left\|\widehat{\boldsymbol{\delta}}_{\mathbb{S}} - \boldsymbol{\delta}_{\mathbb{S}}^\star\right\|_1 + \left\|\widehat{\boldsymbol{\delta}}_{\mathbb{S}^c}\right\|_1,
\end{aligned}
\tag{42}
$$

where $\mathbb{S}^c$ refers to the set of indices of a vector except for $\mathbb{S} = supp(\boldsymbol{\theta}^\star - \boldsymbol{\theta}_{\mathrm{s}}^\star)$. Similarly, noting that $\|\boldsymbol{\delta}_{\mathbb{S}^c}^\star\|_1 = 0$ by definition of $\mathbb{S}$, we have

$$
\begin{aligned}
\left\|\widetilde{\boldsymbol{\delta}}\right\|_1 &= \|\boldsymbol{\delta}^\star - \boldsymbol{\nu}\|_1 \\
&\leq \|\boldsymbol{\delta}_{\mathbb{S}}^\star\|_1 + \|\boldsymbol{\nu}\|_1,
\end{aligned}
\tag{43}
$$

where $\boldsymbol{\nu} = \widehat{\boldsymbol{\theta}}_{\mathrm{s}} - \boldsymbol{\theta}_{\mathrm{s}}^\star$. Combining Equation 42 and Equation 43 and substituting into Equation 41, it further implies

$$
\min_{\Delta = t} \widehat{\mathcal{R}}_\tau(\widehat{\boldsymbol{\delta}}) - \widehat{\mathcal{R}}_\tau(\widetilde{\boldsymbol{\delta}}) - \lambda\left\|\widehat{\boldsymbol{\delta}}_{\mathbb{S}} - \boldsymbol{\delta}_{\mathbb{S}}^\star\right\|_1 + \lambda\left\|\widehat{\boldsymbol{\delta}}_{\mathbb{S}^c}\right\|_1 - \lambda\|\boldsymbol{\nu}\|_1 < 0.
\tag{44}
$$

Furthermore, under the event $\mathcal{J}_{(\widehat{\boldsymbol{\delta}}, \widetilde{\boldsymbol{\delta}})}$ holds and $\lambda \geq 2\lambda_0$, we can replace the $\widehat{\mathcal{R}}_\tau(\cdot)$ with $\mathcal{R}_\tau(\cdot)$, we have

$$
\min_{\Delta = t} \mathcal{R}_\tau(\widehat{\boldsymbol{\delta}}) - \mathcal{R}_\tau(\widetilde{\boldsymbol{\delta}}) - \frac{1}{2}\lambda t - \lambda\left\|\widehat{\boldsymbol{\delta}}_{\mathbb{S}} - \boldsymbol{\delta}_{\mathbb{S}}^\star\right\|_1 + \lambda\left\|\widehat{\boldsymbol{\delta}}_{\mathbb{S}^c}\right\|_1 - \lambda\|\boldsymbol{\nu}\|_1 < 0.
\tag{45}
$$

According to Knight (1998), for any two scalars $w$ and $v$, we have

$$\rho_\tau(w - v) - \rho_\tau(w) = -v(\tau - \mathbf{1}\{w \leq 0\}) + \int_0^v (\mathbf{1}\{w \leq z\} - \mathbf{1}\{w \leq 0\})dz. \qquad (46)$$

Using Equation 46 with $w = y - \boldsymbol{x}'(\widetilde{\boldsymbol{\delta}} + \widehat{\boldsymbol{\theta}}_{\mathrm{s}})$ and $v = \boldsymbol{x}'(\widetilde{\boldsymbol{\delta}} - \widehat{\boldsymbol{\delta}})$, and taking the expectation of both side in Equation 46, we conclude $\mathbb{E}[-v(u - \mathbf{1}\{w \leq 0\})] = 0$. Let $F_{y|x}$ denote the conditional distribution of $y$ given target sample $\boldsymbol{x}$. Using the law of iterated expectations and the expansion of the mean value, we obtain for $\tilde{z}_{x,z} \in [0, z]$,

$$\mathcal{R}_\tau(\widehat{\boldsymbol{\delta}}) - \mathcal{R}_\tau(\widetilde{\boldsymbol{\delta}}) = \mathbb{E}\left[\int_0^{\boldsymbol{x}'(\widetilde{\boldsymbol{\delta}} - \widehat{\boldsymbol{\delta}})} F_{y|x}\left(\boldsymbol{x}'(\widehat{\boldsymbol{\delta}} + \widehat{\boldsymbol{\theta}}_{\mathrm{s}}) + z\right) - F_{y|x}\left(\boldsymbol{x}'(\widehat{\boldsymbol{\delta}} + \widehat{\boldsymbol{\theta}}_{\mathrm{s}})\right) dz\right]$$

$$= \mathbb{E}\left[\int_0^{\boldsymbol{x}'(\widetilde{\boldsymbol{\delta}} - \widehat{\boldsymbol{\delta}})} z f_{y|x}\left(\boldsymbol{x}'(\widehat{\boldsymbol{\delta}} + \widehat{\boldsymbol{\theta}}_{\mathrm{s}})\right) + \frac{z^2}{2} f'_{y|x}\left(\boldsymbol{x}'(\widehat{\boldsymbol{\delta}} + \widehat{\boldsymbol{\theta}}_{\mathrm{s}}) + \tilde{z}_{x,z}\right) dz\right] \qquad (47)$$

$$\geq \frac{1}{2}\underline{f}\left\|\boldsymbol{\Sigma}^{1/2}(\widetilde{\boldsymbol{\delta}} - \widehat{\boldsymbol{\delta}})\right\|_2^2 - \frac{1}{6}\bar{f}'\mathbb{E}\left[\left|\boldsymbol{x}'(\widetilde{\boldsymbol{\delta}} - \widehat{\boldsymbol{\delta}})\right|^3\right],$$

Under the growth condition 39 in the lemma, which implies that

$$\frac{1}{2}\underline{f}\mathbb{E}\left[\left|\boldsymbol{x}'(\widetilde{\boldsymbol{\delta}} - \widehat{\boldsymbol{\delta}})\right|^2\right] > \frac{1}{3}\bar{f}'\mathbb{E}\left[\left|\boldsymbol{x}'(\widetilde{\boldsymbol{\delta}} - \widehat{\boldsymbol{\delta}})\right|^3\right]. \qquad (48)$$

Applying the result of Equation 47 and Equation 48, we can rewrite Equation 45 as

$$\min_{\Delta = t} \frac{1}{4}\underline{f}\left\|\boldsymbol{\Sigma}^{1/2}(\widetilde{\boldsymbol{\delta}} - \widehat{\boldsymbol{\delta}})\right\|_2^2 - \frac{1}{2}\lambda t - \lambda\left\|\widehat{\boldsymbol{\delta}}_{\mathbb{S}} - \boldsymbol{\delta}_{\mathbb{S}}^\star\right\|_1 + \lambda\left\|\widehat{\boldsymbol{\delta}}_{\mathbb{S}^c}\right\|_1 - \lambda\|\boldsymbol{\nu}\|_1 < 0. \qquad (49)$$

Then, we want to apply the assumption 3.3 to $\boldsymbol{\delta} = \widehat{\boldsymbol{\delta}} - \boldsymbol{\delta}^\star$ to bound $\|\widehat{\boldsymbol{\delta}}_{\mathbb{S}} - \boldsymbol{\delta}_{\mathbb{S}}^\star\|_1$, this require the $\widehat{\boldsymbol{\delta}} - \boldsymbol{\delta}^\star$ in the restricted set, which may not always hold in general. To address this, we perform case analysis based on whether $\|\boldsymbol{\nu}\|_1 \leq \|\widehat{\boldsymbol{\delta}}_{\mathbb{S}} - \boldsymbol{\delta}_{\mathbb{S}}^\star\|_1$. We will show that when $\|\boldsymbol{\nu}\|_1 \leq \|\widehat{\boldsymbol{\delta}}_{\mathbb{S}} - \boldsymbol{\delta}_{\mathbb{S}}^\star\|_1$ holds ture, assumption 3.3 to $\boldsymbol{\delta} = \widehat{\boldsymbol{\delta}} - \boldsymbol{\delta}^\star$ can be used to finish our proof, while $\|\boldsymbol{\nu}\|_1 > \|\widehat{\boldsymbol{\delta}}_{\mathbb{S}} - \boldsymbol{\delta}_{\mathbb{S}}^\star\|_1$ also provide a control over the error of the estimator.

First, we discuss the case when $\|\boldsymbol{\nu}\|_1 \leq \|\widehat{\boldsymbol{\delta}}_{\mathbb{S}} - \boldsymbol{\delta}_{\mathbb{S}}^\star\|_1$. According to Equation 24, there exist at least one $\widehat{\boldsymbol{\delta}}$ such that $\Delta = t$ and

$$\frac{1}{4}\underline{f}\left\|\boldsymbol{\Sigma}^{1/2}(\widetilde{\boldsymbol{\delta}} - \widehat{\boldsymbol{\delta}})\right\|_2^2 - \frac{1}{2}\lambda\left\|\widehat{\boldsymbol{\delta}} - \widetilde{\boldsymbol{\delta}}\right\|_1 - \lambda\left\|\widehat{\boldsymbol{\delta}}_{\mathbb{S}} - \boldsymbol{\delta}_{\mathbb{S}}^\star\right\|_1 + \lambda\left\|\widehat{\boldsymbol{\delta}}_{\mathbb{S}^c}\right\|_1 - \lambda\|\boldsymbol{\nu}\|_1 < 0 \qquad (50)$$

holds true. Rearrange the inequality by moving the negative term to the right hand side, we obtain

$$\frac{1}{4}\underline{f}\left\|\boldsymbol{\Sigma}^{1/2}(\widetilde{\boldsymbol{\delta}} - \widehat{\boldsymbol{\delta}})\right\|_2^2 + \lambda\left\|\widehat{\boldsymbol{\delta}}_{\mathbb{S}^c}\right\|_1 < \frac{1}{2}\lambda\left\|\widehat{\boldsymbol{\delta}} - \widetilde{\boldsymbol{\delta}}\right\|_1 + \lambda\left\|\widehat{\boldsymbol{\delta}}_{\mathbb{S}} - \boldsymbol{\delta}_{\mathbb{S}}^\star\right\|_1 + \lambda\|\boldsymbol{\nu}\|_1. \qquad (51)$$

Observing that

$$\left\|\widehat{\boldsymbol{\delta}} - \widetilde{\boldsymbol{\delta}}\right\|_1 = \left\|\widehat{\boldsymbol{\delta}} - \boldsymbol{\delta}^\star + \boldsymbol{\nu}\right\|_1$$
$$\leq \left\|\widehat{\boldsymbol{\delta}}_{\mathbb{S}} - \boldsymbol{\delta}_{\mathbb{S}}^\star\right\|_1 + \left\|\widehat{\boldsymbol{\delta}}_{\mathbb{S}^c}\right\|_1 + \|\boldsymbol{\nu}\|_1, \qquad (52)$$

so we can further simplify Equation 51 to

$$\frac{1}{4}\underline{f}\left\|\boldsymbol{\Sigma}^{1/2}(\widetilde{\boldsymbol{\delta}} - \widehat{\boldsymbol{\delta}})\right\|_2^2 + \frac{\lambda}{2}\left\|\widehat{\boldsymbol{\delta}}_{\mathbb{S}^c}\right\|_1 < \frac{3\lambda}{2}\left\|\widehat{\boldsymbol{\delta}}_{\mathbb{S}} - \boldsymbol{\delta}_{\mathbb{S}}^\star\right\|_1 + \frac{3\lambda}{2}\|\boldsymbol{\nu}\|_1$$
$$\leq 3\lambda\left\|\widehat{\boldsymbol{\delta}}_{\mathbb{S}} - \boldsymbol{\delta}_{\mathbb{S}}^\star\right\|_1, \qquad (53)$$

where the second inequality holds by $\|\boldsymbol{\nu}\|_1 \leq \|\widehat{\boldsymbol{\delta}}_{\mathbb{S}} - \boldsymbol{\delta}_{\mathbb{S}}^\star\|_1$. Dropping the first non-negative term on the left hand side, we can observe that $\widehat{\boldsymbol{\delta}} - \boldsymbol{\delta}^\star$ meets the definition of the restricted set $\mathbb{A}(\mathbb{S}, \alpha)$ with $\alpha = 6$ and $\mathbb{S} = supp(\boldsymbol{\theta}^\star - \boldsymbol{\theta}_{\mathrm{s}}^\star)$. That is

$$\left\|\widehat{\boldsymbol{\delta}}_{\mathbb{S}^c} - \boldsymbol{\delta}_{\mathbb{S}^c}^\star\right\|_1 \leq 6\left\|\widehat{\boldsymbol{\delta}}_{\mathbb{S}} - \boldsymbol{\delta}_{\mathbb{S}}^\star\right\|_1.$$

and we can apply the assumption 3.3 to process. This yields

$$\lambda \left\| \widehat{\boldsymbol{\delta}}_{\mathbb{S}} - \boldsymbol{\delta}_{\mathbb{S}}^{\star} \right\|_1 \leq \frac{\lambda \sqrt{s}}{\kappa} \left\| \boldsymbol{\Sigma}^{1/2}(\widehat{\boldsymbol{\delta}} - \boldsymbol{\delta}^{\star}) \right\|_2 \tag{54}$$

$$\leq \frac{1}{8}\underline{f} \left\| \boldsymbol{\Sigma}^{1/2}(\widehat{\boldsymbol{\delta}} - \boldsymbol{\delta}^{\star}) \right\|_2^2 + \frac{2\lambda^2 s}{\underline{f}\kappa^2} \tag{55}$$

$$\leq \frac{1}{4}\underline{f} \left\| \boldsymbol{\Sigma}^{1/2}(\widehat{\boldsymbol{\delta}} - \widetilde{\boldsymbol{\delta}}) \right\|_2^2 + \frac{1}{4}\underline{f} \left\| \boldsymbol{\Sigma}^{1/2}\boldsymbol{\nu} \right\|_2^2 + \frac{2\lambda^2 s}{\underline{f}\kappa^2}, \tag{56}$$

where the second inequality holds since $ab \leq a^2/4 + b^2$ and the last inequality holds by $(a+b)^2 \leq 2a^2 + 2b^2$. Moreover, note that the variance matrix for target data $\boldsymbol{\Sigma} \in \mathbb{R}^{d \times d}$ is a square matrix, we have

$$\begin{aligned} \left\| \boldsymbol{\Sigma}^{1/2}\boldsymbol{\nu} \right\|_2 &\leq \left\| \boldsymbol{\Sigma}^{1/2} \right\|_2 \|\boldsymbol{\nu}\|_2 \\ &\leq \sqrt{\mathrm{tr}\left(\boldsymbol{\Sigma}\right)} \|\boldsymbol{\nu}\|_2 \\ &= \sqrt{d} \|\boldsymbol{\nu}\|_2 \,, \end{aligned} \tag{57}$$

where the first inequality holds by the definition of matrix norm, and the second inequality holds by the Jensen's inequality, and the last equality holds since we standardize the covariance matrices in assumption 3.1, which implies that sum of the diagonal elements of $\boldsymbol{\Sigma}$ equals to $d$. According to Equation 56 and Equation 57, we observe that

$$\lambda \left\| \widehat{\boldsymbol{\delta}}_{\mathbb{S}} - \boldsymbol{\delta}_{\mathbb{S}}^{\star} \right\|_1 \leq \frac{1}{4}\underline{f} \left\| \boldsymbol{\Sigma}^{1/2}(\widehat{\boldsymbol{\delta}} - \widetilde{\boldsymbol{\delta}}) \right\|_2^2 + \frac{1}{4}\underline{f}d \|\boldsymbol{\nu}\|_2^2 + \frac{2\lambda^2 s}{\underline{f}\kappa^2}. \tag{58}$$

Using these facts in Equation 52 and Equation 58 to bound the $\|\widehat{\boldsymbol{\delta}}_{\mathbb{S}^c}\|_1$ and $\|\widehat{\boldsymbol{\delta}}_{\mathbb{S}} - \boldsymbol{\delta}_{\mathbb{S}}^{\star}\|_1$ respectively in Equation 49, we obtain such relation

$$\lambda t < \underline{f}d \|\boldsymbol{\nu}\|_2^2 + 4\lambda \|\boldsymbol{\nu}\|_1 + \frac{8\lambda^2 s}{\underline{f}\kappa^2}. \tag{59}$$

which is impossible according to the value of $t$.

Lastly, we remain to discuss the case $\|\widehat{\boldsymbol{\delta}}_{\mathbb{S}} - \boldsymbol{\delta}_{\mathbb{S}}^{\star}\|_1 \leq \|\boldsymbol{\nu}\|_1$. Using the intermediate result in Equation 52 to replace the $\|\widehat{\boldsymbol{\delta}}_{\mathbb{S}^c}\|_1$ in Equation 49 again, we get

$$\min_{\Delta=t} \frac{1}{4}\underline{f} \left\| \boldsymbol{\Sigma}^{1/2}(\widetilde{\boldsymbol{\delta}} - \widehat{\boldsymbol{\delta}}) \right\|_2^2 + \frac{1}{2}\lambda t - 2\lambda \left\| \widehat{\boldsymbol{\delta}}_{\mathbb{S}} - \boldsymbol{\delta}_{\mathbb{S}}^{\star} \right\|_1 - 2\lambda \|\boldsymbol{\nu}\|_1 < 0. \tag{60}$$

Applying $\|\widehat{\boldsymbol{\delta}}_{\mathbb{S}} - \boldsymbol{\delta}_{\mathbb{S}}^{\star}\|_1 \leq \|\boldsymbol{\nu}\|_1$, we have

$$\min_{\Delta=t} \frac{1}{4}\underline{f} \left\| \boldsymbol{\Sigma}^{1/2}(\widetilde{\boldsymbol{\delta}} - \widehat{\boldsymbol{\delta}}) \right\|_2^2 + \frac{1}{2}\lambda t - 4\lambda \|\boldsymbol{\nu}\|_1 < 0. \tag{61}$$

Dropping the first non-negative term $1/4 \cdot \underline{f}\|\boldsymbol{\Sigma}^{1/2}(\widetilde{\boldsymbol{\delta}} - \widehat{\boldsymbol{\delta}})\|_2^2$, we obtain such relation

$$\lambda t < 8\lambda \|\boldsymbol{\nu}\|_1. \tag{62}$$

is impossible according to the value of $t$. $\qquad\square$

### A.3 PROOF OF THEOREM 3.5 AND COROLLARY

*Proof.* By Lemma A.2 and Lemma A.3, we have with $\lambda \geq 2\lambda_0$,

$$\begin{aligned} \mathrm{P}&\left( \left\| \widehat{\boldsymbol{\theta}} - \boldsymbol{\theta}^{\star} \right\|_1 \leq \frac{\underline{f}d}{\lambda} \|\boldsymbol{\nu}\|_2^2 + 8 \|\boldsymbol{\nu}\|_1 + \frac{8\lambda s}{\underline{f}\kappa^2} \right) \\ &\geq \mathrm{P}(\mathcal{J}_{(\widehat{\boldsymbol{\delta}}, \widetilde{\boldsymbol{\delta}})}) \\ &\geq 1 - 8d \cdot \exp\left( -\frac{\lambda_0^2 \cdot n}{32 L_\tau^2} \right). \end{aligned} \tag{63}$$

Recall that $\lambda_0$ is theoretical coefficient about event $\mathcal{J}_{(\widehat{\boldsymbol{\delta}},\widetilde{\boldsymbol{\delta}})}$, we can choose to optimize our bound. Thus, by choosing $\lambda_0 = \sqrt{32L_\tau^2(\log(8d)+u)/n}$ for any $u > 0$, we have with probability at least $1 - e^{-u}$,

$$\left\|\widehat{\boldsymbol{\theta}} - \boldsymbol{\theta}^\star\right\|_1 \leq \frac{\underline{f}d}{\lambda}\|\boldsymbol{\nu}\|_2^2 + 8\|\boldsymbol{\nu}\|_1 + \frac{8\lambda s}{\underline{f}\kappa^2}. \tag{64}$$

By inspection, plugging in

$$\lambda^\star = C\max\left\{\sqrt{\frac{128L_\tau^2(\log(8d)+u)}{n}}, d\|\boldsymbol{\nu}\|_2\right\},$$

with tuning parameter $C > 1$. We obtain with probability at least $1 - e^{-u}$,

$$\left\|\widehat{\boldsymbol{\theta}} - \boldsymbol{\theta}^\star\right\|_1 \leq \mathcal{O}\left(\max\left\{s\sqrt{\frac{\log(d)+u}{n}}, ds\|\boldsymbol{\nu}\|_2\right\}\right). \tag{65}$$

Next, we remain to derive the expected out-of-sample prediction error. For convenience let

$$w := \frac{\underline{f}d}{\lambda}\|\boldsymbol{\nu}\|_2^2 + 8\|\boldsymbol{\nu}\|_1 + \frac{8\lambda s}{\underline{f}\kappa^2}.$$

By Hölder's inequality, we have

$$\mathbb{E}\left[\left\|\boldsymbol{x}'(\widehat{\boldsymbol{\theta}}^{\mathrm{CLIP}} - \boldsymbol{\theta}^\star)\right\|_1\right] \leq \mathbb{E}\left[\left\|\widehat{\boldsymbol{\theta}}^{\mathrm{CLIP}} - \boldsymbol{\theta}^\star\right\|_1\right] \cdot \|\boldsymbol{x}\|_\infty, \tag{66}$$

where $\|\boldsymbol{x}\|_\infty$ is the largest magnitude among each element of vector $\boldsymbol{x}$. To bound $\mathbb{E}[\|\widehat{\boldsymbol{\theta}}^{\mathrm{CLIP}} - \boldsymbol{\theta}^\star\|_1]$, We can proceed by conducting some case analysis. Recall that the definition of the event $\mathcal{J}_{(\widehat{\boldsymbol{\delta}},\widetilde{\boldsymbol{\delta}})}$ is

$$\mathcal{J}_{(\widehat{\boldsymbol{\delta}},\widetilde{\boldsymbol{\delta}})} := \left\{\sup_{\|\widehat{\boldsymbol{\delta}}-\widetilde{\boldsymbol{\delta}}\|_1\leq t}\left\|\widehat{\mathcal{R}}_\tau(\widehat{\boldsymbol{\delta}}) - \widehat{\mathcal{R}}_\tau(\widetilde{\boldsymbol{\delta}}) - (\mathcal{R}_\tau(\widehat{\boldsymbol{\delta}}) - \mathcal{R}_\tau(\widetilde{\boldsymbol{\delta}}))\right\|_1 \leq \lambda_0 t\right\}.$$

That yields

$$\mathbb{E}\left[\left\|\widehat{\boldsymbol{\theta}}^{\mathrm{CLIP}} - \boldsymbol{\theta}^\star\right\|_1\right] = \mathbb{E}\left[\left\|\widehat{\boldsymbol{\theta}}^{\mathrm{CLIP}} - \boldsymbol{\theta}^\star\right\|_1 \,\Big|\, \mathcal{J}_{(\widehat{\boldsymbol{\delta}},\widetilde{\boldsymbol{\delta}})}\right] \cdot \mathrm{P}[\mathcal{J}_{(\widehat{\boldsymbol{\delta}},\widetilde{\boldsymbol{\delta}})}] + \\ \mathbb{E}\left[\left\|\widehat{\boldsymbol{\theta}}^{\mathrm{CLIP}} - \boldsymbol{\theta}^\star\right\|_1 \,\Big|\, \mathcal{J}_{(\widehat{\boldsymbol{\delta}},\widetilde{\boldsymbol{\delta}})}^C\right] \cdot \mathrm{P}[\mathcal{J}_{(\widehat{\boldsymbol{\delta}},\widetilde{\boldsymbol{\delta}})}^C]. \tag{67}$$

To bound the first expectation on the right-hand side of Equation 67, we further define a new event

$$\mathcal{B} = \left(\left\|\widehat{\boldsymbol{\theta}}\right\|_1 \leq 2b\right).$$

Recall the definition of $\widehat{\boldsymbol{\theta}}^{\mathrm{CLIP}}$, we know that $\widehat{\boldsymbol{\theta}}^{\mathrm{CLIP}} = \widehat{\boldsymbol{\theta}}$ when $\mathcal{B}$ holds, and $\widehat{\boldsymbol{\theta}}^{\mathrm{CLIP}} = 0$ otherwise. Then,

$$\mathbb{E}\left[\left\|\widehat{\boldsymbol{\theta}}^{\mathrm{CLIP}} - \boldsymbol{\theta}^\star\right\|_1 \,\Big|\, \mathcal{J}_{(\widehat{\boldsymbol{\delta}},\widetilde{\boldsymbol{\delta}})}\right]$$
$$= \mathbb{E}\left[\left\|\widehat{\boldsymbol{\theta}}^{\mathrm{CLIP}} - \boldsymbol{\theta}^\star\right\|_1 \,\Big|\, \mathcal{B}\cap\mathcal{J}_{(\widehat{\boldsymbol{\delta}},\widetilde{\boldsymbol{\delta}})}\right]\cdot\mathrm{P}[\mathcal{B}] + \mathbb{E}\left[\left\|\widehat{\boldsymbol{\theta}}^{\mathrm{CLIP}} - \boldsymbol{\theta}^\star\right\|_1 \,\Big|\, \mathcal{B}^C\cap\mathcal{J}_{(\widehat{\boldsymbol{\delta}},\widetilde{\boldsymbol{\delta}})}\right]\cdot\mathrm{P}\left[\mathcal{B}^C\right] \tag{68}$$
$$= \mathbb{E}\left[\left\|\widehat{\boldsymbol{\theta}} - \boldsymbol{\theta}^\star\right\|_1 \,\Big|\, \mathcal{B}\cap\mathcal{J}_{(\widehat{\boldsymbol{\delta}},\widetilde{\boldsymbol{\delta}})}\right]\cdot\mathrm{P}[\mathcal{B}] + \mathbb{E}\left[\|\boldsymbol{\theta}^\star\|_1 \,\Big|\, \mathcal{B}^C\cap\mathcal{J}_{(\widehat{\boldsymbol{\delta}},\widetilde{\boldsymbol{\delta}})}\right]\cdot\mathrm{P}\left[\mathcal{B}^C\right].$$

Now, note that on the event $\mathcal{B}^C\cap\mathcal{J}_{(\widehat{\boldsymbol{\delta}},\widetilde{\boldsymbol{\delta}})}$, we have both that

$$\left\|\widehat{\boldsymbol{\theta}} - \boldsymbol{\theta}^\star\right\|_1 \leq w, \quad \left\|\widehat{\boldsymbol{\theta}}\right\|_1 \geq 2b \geq 2\|\boldsymbol{\theta}^\star\|_1.$$

Combining these facts together, we have on the event $\mathcal{B}^C\cap\mathcal{J}_{(\widehat{\boldsymbol{\delta}},\widetilde{\boldsymbol{\delta}})}$,

$$\|\boldsymbol{\theta}^\star\|_1 \leq \left\|\widehat{\boldsymbol{\theta}}\right\|_1 - \|\boldsymbol{\theta}^\star\|_1 \leq \left\|\widehat{\boldsymbol{\theta}} - \boldsymbol{\theta}^\star\right\|_1 \leq w,$$

always holds using the triangle inequality. Thus first expectation on the right-hand side of Equation 67 can obtain

$$
\begin{aligned}
\mathbb{E}\left[\left\|\widehat{\boldsymbol{\theta}}^{\mathrm{CLIP}} - \boldsymbol{\theta}^{\star}\right\|_1 \,\Big|\, \mathcal{J}_{(\widehat{\boldsymbol{\delta}}, \widetilde{\boldsymbol{\delta}})}\right] &\leq w \cdot \mathrm{P}[\mathcal{B}] + \mathbb{E}\left[\left\|\boldsymbol{\theta}^{\star}\right\|_1 \,\Big|\, \mathcal{B}^C \cap \mathcal{J}_{(\widehat{\boldsymbol{\delta}}, \widetilde{\boldsymbol{\delta}})}\right] \cdot \mathrm{P}\left[\mathcal{B}^C\right] \\
&\leq w \cdot \mathrm{P}[\mathcal{B}] + w \cdot \mathrm{P}\left[\mathcal{B}^C\right] \\
&= w.
\end{aligned}
\tag{69}
$$

Next, we continue to bound the second expectation on the right-hand side of Equation 67. Regardless of the events $\mathcal{J}_{(\widehat{\boldsymbol{\delta}}, \widetilde{\boldsymbol{\delta}})}$ and $\mathcal{B}$, using the triangle inequality, we have

$$
\left\|\widehat{\boldsymbol{\theta}}^{\mathrm{CLIP}} - \boldsymbol{\theta}^{\star}\right\|_1 \leq \left\|\widehat{\boldsymbol{\theta}}^{\mathrm{CLIP}}\right\|_1 + \left\|\boldsymbol{\theta}^{\star}\right\|_1 \leq 3b.
\tag{70}
$$

Combining Equation 67, Equation 69 and Equation 70, we have

$$
\begin{aligned}
\mathbb{E}\left[\left\|\widehat{\boldsymbol{\theta}}^{\mathrm{CLIP}} - \boldsymbol{\theta}^{\star}\right\|_1\right] &= w \cdot \mathrm{P}[\mathcal{J}_{(\widehat{\boldsymbol{\delta}}, \widetilde{\boldsymbol{\delta}})}] + 3b \cdot \mathrm{P}[\mathcal{J}^C_{(\widehat{\boldsymbol{\delta}}, \widetilde{\boldsymbol{\delta}})}] \\
&\leq w + 3b \cdot \mathrm{P}[\mathcal{J}^C_{(\widehat{\boldsymbol{\delta}}, \widetilde{\boldsymbol{\delta}})}] \\
&\leq w + 24bd \cdot \exp\left(-\frac{\lambda^2 \cdot n}{128 L_\tau^2}\right).
\end{aligned}
\tag{71}
$$

where the last inequality holds by using the result in Lemma A.2 with $\lambda_0 = \lambda/2$. Taking the regularization hyperparameter $\lambda$ to be

$$
\lambda^{\star} = C \max\left\{\sqrt{\frac{128 L_\tau^2 \log(24bdn)}{n}},\, d\left\|\boldsymbol{\nu}\right\|_2\right\},
$$

which yields expected out-of-sample prediction error for any new coming input $\boldsymbol{x}$ as

$$
\mathbb{E}\left[\left\|\boldsymbol{x}'\widehat{\boldsymbol{\theta}} - \boldsymbol{x}'\boldsymbol{\theta}^{\star}\right\|_1\right] \leq \mathcal{O}\left(\max\left\{\frac{s\left\|\boldsymbol{x}\right\|_\infty}{\sqrt{n}} \log(dn), ds\left\|\boldsymbol{\nu}\right\|_2 \left\|\boldsymbol{x}\right\|_\infty\right\}\right).
$$

$\square$

# B  EXPERIMENT DETAILS

## B.1  PRACTICAL IMPLEMENTATION

We evaluate the adaptation efficiency of QAdapter by comparing it with baselines, including DT, Zero-shot, Average, and Lasso, in simulation. Our transfer learning algorithm is divided into two main steps:

1. **Pretraining with Source Data**: In the first step, we pretrain the model using the source data. In the simulation, our pretrained model is trained as follows:

$$
\widehat{\boldsymbol{\theta}}_{\mathrm{s}} = \arg\min_{\boldsymbol{\theta}} \frac{1}{n} \sum_{i=0}^{n} \rho_\tau(y_i - \boldsymbol{x}_i'\boldsymbol{\theta}), \quad (\boldsymbol{x}_i, y) \sim \mathcal{D}_{\mathrm{s}}.
\tag{72}
$$

2. **Adapting for Downstream Tasks**: In this step, we train additional task-specific parameters to adapt to downstream tasks by $\mathcal{D}$.

Our code is implemented in Python, and we optimize all baseline objective functions using CVXPY: an open-source Python package for convex optimization problems. We run 100 seeds for each experiment and record the mean of MSE and quantile loss. We plot the results under different similarity coefficients $|\mathbb{S}| = \{0, 5, 10, 20, 30, \ldots, d\}$, $\lambda = \{0, 0.05, 0.1, \ldots, 0.4\}$, and $n = \{150, 200, 300, \ldots, 1000\}$ on the x-axis of Figure 1.

### B.2 BASELINES

For **DT**, we directly train the target estimator using target data $\mathcal{D}$:

$$\widehat{\boldsymbol{\theta}}_{DT} = \arg\min_{\boldsymbol{\theta}} \frac{1}{n} \sum_{i=0}^{n} \left( y - \boldsymbol{x}'(\widehat{\boldsymbol{\theta}}_{\mathrm{s}} + \boldsymbol{\delta}) \right)^2. \tag{73}$$

We then evaluate the performance on test data with $\widehat{\boldsymbol{\theta}}_{DT}$, without any transfer learning step.

For **Zero-shot**, we directly evaluate the pretrained model $\widehat{\boldsymbol{\theta}}_{\mathrm{s}}$ on test data without additional parameter updates. The pretrained $\widehat{\boldsymbol{\theta}}_{\mathrm{s}}$ comes from Equation 72.

For **QAdapter**, we optimize Equation 10 to obtain the adapter and perform inference on test data using $\widehat{\boldsymbol{\delta}} + \widehat{\boldsymbol{\theta}}_{\mathrm{s}}$. We set $\tau = 0.5$ by default and use $\lambda = 0.01$ for quantile adaptation in Figure 1, and $\lambda = 0.1$ for the extreme value prediction task.

For **LAdapter**, we train with the lasso objective:

$$\widehat{\boldsymbol{\delta}}_L = \arg\min_{\boldsymbol{\delta}} \frac{1}{n} \sum_{i=0}^{n} \left( y - \boldsymbol{x}'(\widehat{\boldsymbol{\theta}}_{\mathrm{s}} + \boldsymbol{\delta}) \right)^2 + \lambda \left\| \boldsymbol{\delta} \right\|_1, \tag{74}$$

where $\lambda$ is the same as for QAdapter. LAdapter performs inference with $\widehat{\boldsymbol{\delta}}_L + \widehat{\boldsymbol{\theta}}_{\mathrm{s}}$.

For **Average**, the estimator is $\alpha_1 \widehat{\boldsymbol{\theta}}_{\mathrm{s}} + (1 - \alpha_1)\widehat{\boldsymbol{\theta}}, \alpha_1 \in (0, 1)$. We choose $\alpha = 0.7$ based on cross-validation methods and perform inference on test data.

### B.3 ADDITIONAL RESULTS

Here we report the quantile loss ($\tau = 0.5$) about the prediction error of adapter in test data in the following figures.

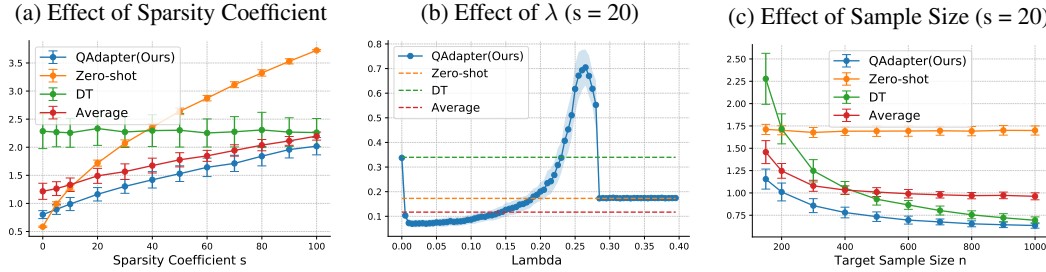

Figure 3: Analysis of various factors affecting model performance, measured using the quantile loss ($\tau = 0.5$) on the y-axis.

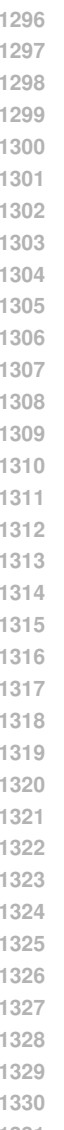

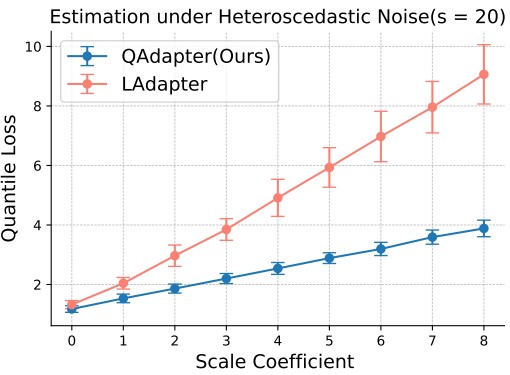

Figure 4: Additional result in heteroscedastic experiment.

