# OpenReview forum: "A Provable Quantile Regression Adapter via Transfer Learning"
_ICLR.cc/2025/Conference — ICLR 2025 Conference Withdrawn Submission_

### Official Review · Reviewer_vkG5 · 2024-11-03

**Soundness:** 2
**Presentation:** 3
**Contribution:** 1
**Rating:** 5
**Confidence:** 3

**Summary:**

This paper studies the linear quantile regression problem under a high-dimensional and transfer learning setting. Specifically, it assumes that the source distribution of $(X, Y)$ satisfies $Q_{\tau}(Y|X) = X^\top \theta_s$ and the target distribution satisfies $Q_{\tau}(Y|X) = X^\top \theta$, where $\theta_s \approx \theta$. Then, under the high-dimension setting with limited data from the target distribution, the paper proposes to learn the $\theta - \theta_s$ term by adding a Lasso regularizer to the original quantile loss and then minimizing the loss.

**Strengths:**

The paper considers a setting that can be summarized as a regularized quantile regression. The problem is interesting by itself. The results in the paper are all supported by rigorous proof.

**Weaknesses:**

The paper aims to extend the Lasso-style analysis to the quantile regression setting, but the proving techniques seem to be relatively standard. As a result, the contribution may not meet the bars for a top Machine Learning conference. Additionally, the scope of the paper seems limited: the theoretical results are restricted to a realizable setting where the conditional quantile is modeled by a linear function, and the experiments are restricted to synthetic data. Furthermore, while the paper mentions transfer learning, the transfer aspect is limited to the assumption of $\theta - \theta_s$ being small. There is no explicit algorithmic design to address or leverage the distribution shift between the source and the target distributions.

**Questions:**

In Line 271, you mentioned "Incorporating this error into the analysis of $\widehat{\theta}$ is nontrivial". Could you please briefly explain where this nontriviality lies?

---

### Official Review · Reviewer_UVRa · 2024-11-03

**Soundness:** 2
**Presentation:** 2
**Contribution:** 2
**Rating:** 3
**Confidence:** 4

**Summary:**

This paper proposes a quantile regression-based transfer learning algorithm designed to transfer knowledge to risk-sensitive downstream tasks. The authors introduce a measure to theoretically quantify the transferability of knowledge and provide statistical guarantees for adaptation efficiency within a linear structural model. They also evaluate the adaptation performance of the algorithm through numerical simulations. The results show the proposed method outperforms the baselines.

**Strengths:**

The proposed research direction—designing transfer learning approaches for risk-sensitive tasks—is an important open problem to solve. The authors have conducted both empirical and theoretical analyses to test and demonstrate the effectiveness of the proposed quantile regression approach in sample-efficient transfer learning.

**Weaknesses:**

The paper's presentation is somewhat misleading. Although the introduction extensively discusses the adapter-tuning strategy in LLMs, the proposed transfer learning approach (linear structural model) cannot be applied to this field, which may confuse readers about the scope and application of the research.

The authors only study on transfer learning under a linear structural model. Both the empirical and theoretical analyses are based on this setting. I am concerned about how well this quantile regression approach will perform in real-world transfer learning tasks. There can be a significant gap when applying the proposed algorithm to transfer learning frameworks that utilize general pre-trained models, which typically involve nonlinear neural network structures such as large language models (LLMs). Additionally, the baselines used for comparison in the study are limited. The experiments are conducted solely on synthetic data, and no real-world transfer learning tasks are included.

**Questions:**

It is beneficial if the authors can test the proposed method on real-world transfer learning tasks using nonlinear (neural network based) pre-trained model (e.g, Mistral-7B, Llama3-8B).

---

### Official Review · Reviewer_5XxH · 2024-11-04

**Soundness:** 3
**Presentation:** 3
**Contribution:** 1
**Rating:** 3
**Confidence:** 4

**Summary:**

This paper studies the adapter-finetuning strategy for the quantile objective. It developed theoretical guarantee under linear models and regularity conditions on the sparsity of the parameter, and uses numerical experiments to demonstrate the effectiveness of the proposed method.

**Strengths:**

The paper is well-written and easy to follow. The quantile objective is well-motivated and the theoretical concepts are well explained.

**Weaknesses:**

My main concern is the technical contribution of the work.

The theoretical analysis:
- To me, the theoretical results of the paper directly apply the existing analyses from high-dimensional statistics and sparse (quantile) regression. The setup of the analysis is separate from the context of adapter fine-tuning. Specifically, if we focus on the $\delta^*$, then all the assumptions in Section 3 are about $\delta^*$, and all the results in Section 3 can be readily implied from the existing tools based on an analysis for $\delta^*$ and completely detached from the adapter fine-tuning context.

The numerical experiments:
- The numerical experiments are in small-sample and low-dimensional regime, which to me, doesn't reflect the application scenarios for adapter fine-tuning. More experiments should be done under a more relevant context such as fine-tuning of larger models.

Minor comments on the technical proofs/analyses:

Overall, the paper might benefit from a more careful proofreading for the technical parts. I might misunderstand some part, but I will list my confusion in below for authors' reference.

- The formula (26) is correct, but might be loose, can you reduce the order of d from 1 to 1/2?
- From formula (30) to (31), you seem to use the inequality e^{|a| + |b|} <= e^{a+b}+e^{-a-b}. But this inequality is not true, for example, when a = 1, b = -1.
- In line 954, I am confused by "Last equality ... column j". I didn't see this assumption in previous text.
- In formula (39), what is q? And why (39) imply formula (48)?
- In line 1031, I believe u should be tau. And why E[-v(tau- 1{w<=0})] = 0? You seem to assume delta^{hat} is unbiased.
- In line 1142, the second part of lambda* should be sqrt{d/s}||v||_2. In formula (65), does the first part come from the plugging in of the first term of lambda*? If so, you seem to be missing a term that comes from underline{f}d/lambda ||v||_2^2. The second part should be sqrt{ds}||v||_2.
- In line 143, j should be in set [d]. Same as line 282.
- In line 145, the sum of i should start at 1, not 0.
- In line 147, only semi-positive matrices have matrix square roots.
- The formula (4) is wrong, the left-hand side should be rho(y-f(x, theta)), or correspondingly change the right-hand side.
- In line 201, I believe the correct one is "to learn theta*", not theta_{s}*.
- Could you give a reference for definition 3.2? In the book by Wainwright (2019), Definition 7.12 does not seem to have the term sqrt(|S|).
- In line 365, I believe the precise expression is "of the order O(d^{1/2}/n_s^{1/2})".
- In line 836, you should modify "The our estimator".
- In line 887, |ab|_1 has no meaning, maybe you should write |a^Tb|.
- In formula (25), the first one should be equality, and the second one should be inequality.
- In line 1030, I believe the correct one is v = x'(delta^{hat} - delta^{tilde}). Same in formula (47).

**Questions:**

See above.

---

### Official Review · Reviewer_GCNU · 2024-11-04

**Soundness:** 2
**Presentation:** 2
**Contribution:** 1
**Rating:** 3
**Confidence:** 3

**Summary:**

This paper proposes a transfer learning approach for quantile regression via adapter-tuning, which enables risk-sensitive adaptation in pretrained models. It introduces a quantile regression adapter that injects sparse, low-rank parameters into a pretrained model to enhance sample efficiency and performance. The proposed quantile regression adapter is equipped with performance guarantee and supported by empirical evidence.

**Strengths:**

The paper is well-presented and easy to follow.

Adapting transfer learning to quantile regression is novel

The paper provides thorough theoretical analysis of proposed method.

**Weaknesses:**

The paper highlights the effectiveness of transfer learning techniques in fine-tuning large pretrained models, but lacking methods and experiments on applying the proposed approach to certain models.

The paper focuses on linear adapter, while this offers theoretical clarity, it is unclear of how to extend the proposed method to non-linear applications. Linear adapter alone has limited applicability.

The left side of equation $(4)$ should be $\rho_{\tau}(y-f(x;\theta))$ instead of $\rho_{\tau}(x)$, as the function $\rho_\tau(\cdot)$ should depend on $y-f(x;\theta)$ rather than directly on $x$ to be consistent with equation $(3)$.

**Questions:**

Please refer to strengths and weaknesses.

---

### Note · Authors · 2024-11-13

**Comment:**

Dear Program Committee and Reviewers,

After careful consideration, we have decided to withdraw our paper from the conference review process. While we appreciate the valuable feedback provided by the reviewers, we believe that addressing these comments thoroughly will require additional time and resources to improve the quality of our work beyond the rebuttal phase.

We would like to express our gratitude for the constructive feedback, which has provided valuable insights into how we can enhance our research. We look forward to carefully implementing these suggestions and potentially resubmitting an improved version to a future venue.

Thank you once again for your time and consideration.

**Withdrawal Confirmation:**

I have read and agree with the venue's withdrawal policy on behalf of myself and my co-authors.